# Exploiting Edge Features in Graph-based Learning with Fused Network Gromov-Wasserstein Distance

**Junjie Yang**                                                   *junjie.yang@telecom-paris.fr*
*LTCI, Télécom Paris*
*IP Paris, France*

**Matthieu Labeau**                                           *matthieu.labeau@telecom-paris.fr*
*LTCI, Télécom Paris*
*IP Paris, France*

**Florence d'Alché-Buc**                                   *florence.dalche@telecom-paris.fr*
*LTCI, Télécom Paris*
*IP Paris, France*

**Reviewed on OpenReview:** *https://openreview.net/forum?id=8uCNtJ2Fmo*

## Abstract

Pairwise comparison of graphs is key to many applications in Machine Learning ranging from clustering, kernel-based classification/regression and more recently supervised graph prediction. Distances between graphs usually rely on informative representations of these structured objects such as bag of substructures or other graph embeddings. A recently popular solution consists in representing graphs as metric measure spaces, allowing to successfully leverage Optimal Transport, which provides meaningful distances allowing to compare them, namely the Gromov-Wasserstein distance and its variant the Fused Gromov-Wasserstein that applies on node attributed graphs. However, this family of distances overlooks edge attributes, which are essential for many structured objects. In this work, we introduce an extension of the Fused Gromov-Wasserstein distance for comparing graphs whose both nodes and edges have features. We propose novel algorithms for distance and barycenter computation. We present a range of studies that illustrate the properties of the proposed distance and empirically demonstrate its effectiveness in supervised graph prediction tasks.

## 1 Introduction

Optimal Transport (OT) (Villani, 2009) has witnessed a growing attention in Machine Learning wherein it has become a key tool to compare probability distributions (Shen et al., 2018). In particular, optimal transport distances exhibit properties, such as their ability to take into account, via ground metrics, the geometry of the sample space and their differentiability, that make them especially suitable as loss functions (see for instance Wasserstein Auto-encoders (Tolstikhin et al., 2018) or Wasserstein Generative Adversarial Neural Networks (Arjovsky et al., 2017)). Particularly of interest here are novel insights that recent works have brought on graph-based learning, benefiting a series of tasks, e.g. graph classification (Vayer et al., 2019), graph matching (Xu et al., 2019), dictionary learning (Vincent-Cuaz et al., 2021) or supervised graph prediction (Brogat-Motte et al., 2022). Based on the representation of a graph as a metric measure space, where the nodes of the graph are considered as the support of the probability measure, OT provides a natural way to compute a meaningful distance between graphs, such as the Gromov-Wasserstein (GW) distance (Mémoli, 2011; Sturm, 2012). The asymmetric structure of directed graphs has led to the generalization of GW via the Network Gromov-Wasserstein (NGW) distance (Chowdhury & Mémoli, 2019) while the Fused Gromov-Wasserstein (FGW) distance (Vayer et al., 2019) was proposed to extend GW to node-labeled graphs. Recently, Barbe et al. (2021) proposed the Diffused Gromov Wasserstein (DFGW) distance

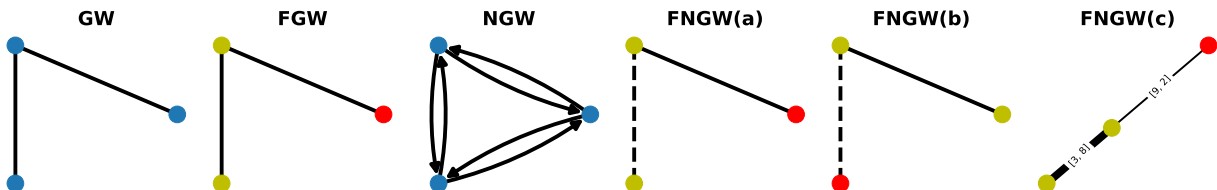

Figure 1: Different types of graphs represented by the GW-based distances

to smooth the node features along the graph structure through a heat diffusion operator which uses the Laplacian kernel. Overall, this new class of distances enjoys useful metric and geodesic properties, allowing for example to derive the minimum path between two structured objects (Sturm, 2012; Vayer et al., 2020).

On the other hand, many machine learning applications require dealing with graphs with complex edge features, such as abstract meaning representation (AMR) (Banarescu et al., 2013) in natural language processing, scene graphs (J. Johnson et al., 2015) in computer vision, or molecules in chemistry (Irwin et al., 2012; Dührkop et al., 2015). For instance, when considering molecule identification tasks, leveraging the Fused Gomov-Wasserstein distance as a loss does not allow to reach the same level of performance than a kernel that considers more complex molecule representations. Dynamic graphs may also require more sophisticated modeling, as they use temporal edge features which continuously evolve over time (Kazemi et al., 2020). There have been several attempts in the literature to obtain meaningful graph representations by including edge features: the two dominant solutions are to use graph kernels and Graph Neural Networks (GNNs). Among graph kernels, the Neighborhood Subgraph Pairwise Distance Kernel (NSPDK) (Costa & Grave, 2010) takes edge labels into consideration to build graph-invariant encodings of subgraphs while more involved graph kernels leverage bags of subgraphs (Grenier et al., 2015). The GNNs belonging to the Message Passing Neural Networks (MPNNs) framework (Simonovsky & Komodakis, 2017; Gilmer et al., 2017; Fey et al., 2018; Corso et al., 2020) may incorporate directly edge features via their aggregation procedure. Some attention-based variants (Veličković et al., 2018; Shi et al., 2021; Brody et al., 2022) leverage edge features to compute the attention weights.

Building on recent computational and theoretical results on GW distances, this paper proposes to represent the edge features of a graph by equipping the original measure space with an additional binary function whose codomain falls in a metric space. This generalization allows us to flexibly include edge features into the computation of a novel fused GW distance while keeping its desirable topological properties. As a consequence, barycenters of edge and node featured graphs can also be computed. These new tools especially unlocks more accurate graph modeling for a wide array of tasks. In this work, we first propose to check the relevance of the novel FNGW distance in distance-based learning where, for example, the distance is used to define a kernel in the input space to solve graph classification tasks. We then target supervised graph prediction applications where the FNGW distance is used as a loss function and the predictive model provides a prediction under the form of a FNGW barycenter.

**Summary of contributions:**

- We propose a new OT-based distance (FNGW), which is a generalization of both the Network Gromov-Wasserstein Distance and the Fused Gromov-Wasserstein Distance to edge-featured graphs.

- We derive an algorithm for the computation of the FNGW distance in the discrete case, along with procedures for barycenter computation, unlocking supervised edge-featured graph prediction.

- We present extensive experiments on node and edge featured graph classification tasks that showcase the relevance of the FNGW distance as a means for input graph comparison.

- We tackle supervised prediction of graphs where the loss function is the FNGW distance. On this task, we consider two real-world problems with and without candidate test sets and extend a method based on graph barycenters, by scaling it to large datasets. We observe a significant increase in performance compared to the use of FGW distance.

**Notations:** For two probability measures $\mu \in \text{Prob}(X_0)$, $\nu \in \text{Prob}(X_1)$ where $X_0$ and $X_1$ are both polish spaces, we note $\Pi(\mu, \nu)$ the set of all couplings of $\mu$ and $\nu$, i.e., the set of probability measures $\pi$ on the product space $X_0 \times X_1$ satisfying $\pi(A_0, X_1) = \mu(A_0)$ and $\pi(X_0, A_1) = \nu(A_1)$ for all $A_0 \in \mathcal{B}(X_0)$, $A_1 \in \mathcal{B}(X_1)$ where $\mathcal{B}(\cdot)$ denotes the Borel $\sigma$-algebra. $\Sigma_n = \{\boldsymbol{h} \in (\mathbb{R}_+)^n \mid \sum_{i=1}^n h_i = 1\}$ is the simplex histogram with $n$ bins. In the case where both $\mu$ and $\nu$ are discrete, i.e. we can write $\mu = \sum_{i=1}^n a_i \delta_{x_i^0}$ and $\nu = \sum_{j=1}^m b_j \delta_{x_j^1}$ with $\boldsymbol{a} \in \Sigma_n$ and $\boldsymbol{b} \in \Sigma_m$, we note $\Pi(\boldsymbol{a}, \boldsymbol{b})$ the set of matrices $\pi \in \mathbb{R}^{n \times m}$ satisfying $\sum_i \pi_{i,j} = b_j$ and $\sum_j \pi_{i,j} = a_i$ despite the abuse of the notation. Here $\delta$ denotes the Dirac measure. We use $\#$ to denote the pushforward operator on measures. We note $\times_n$ the $n$-mode tensor-matrix product. Given a tensor $X \in \mathbb{R}^{I_1 \times I_2 \cdots \times I_N}$ and a matrix $A \in \mathbb{R}^{J \times I_n}$, $X \times_n A$ gives a tensor of shape $I_1 \times \cdots \times I_{n-1} \times J \times I_{n+1} \times \cdots \times I_N$ where $(X \times_n A)(i_1, \cdots, i_{n-1}, j, i_{n+1}, \cdots, i_N) = \sum_{i_n=1}^{I_n} X(i_1, \cdots, i_n, \cdots, i_N) A(j, i_n)$. We note $\mathcal{I}_{N \times M}$ a tensor of shape $N \times N \times M$ with $\mathcal{I}_{N \times M}(n_1, n_2, m) = \hat{\delta}_{n_1 n_2}$ where $\hat{\delta}$ is the Kronecker delta function. $A^\dagger$ denotes the Moore-Penrose pseudoinverse of a matrix $A$.

## 2 Fused Network Gromov-Wasserstein Distance and Barycenter

In this section, we first give the discrete definition of the Fused Network Gromov-Wasserstein Distance, designed for the representation of edge-featured graphs. We present our method for computing the distance, along with its constraints. We also introduce a general form of our distance. We end with an algorithm for computing a FNGW-based barycenter. Proofs of propositions and theorems are given in Appendix A.

### 2.1 Fused Network Gromov-Wasserstein Distance and Computation

Along the paper, we use the following definition for describing a node and edge featured graph.

**Definition 2.1** (Node and Edge Featured Graph). A node and edge featured graph is a quadruple of the form $(F, A, E, \boldsymbol{p})$ where $F \in \Psi^m$ is a tuple of points valued in a metric space $(\Psi, d_\Psi)$, $A \in \mathbb{R}^{m \times m}$ is a real-valued matrix, $E \in \Omega^{m \times m}$ is a tuple of points valued in a metric space $(\Omega, d_\Omega)$ and $\boldsymbol{p} \in \Sigma_m$ is a simplex histogram. We denote $\mathcal{G}$ as a set of such quadruples.

We illustrate this definition with the following example.

**Example 2.2** (Node and Edge Featured Graph). Consider the graph (a) in Figure 1 as an illustration. Here, the space $\Psi = \{\text{red}, \text{yellow}\}$ can be chosen to be the node-color space, while $\Omega = \{\text{solid}, \text{dashed}, \text{non-edge}\}$ can be designated as the edge-type space. The elements in both spaces are encoded using a one-hot encoding scheme equipped with Euclidean distance $d_\Psi$ and $d_\Omega$. Consequently, we can represent the graph with:

$$F = \begin{bmatrix} [1,0] \\ [1,0] \\ [0,1] \end{bmatrix}, \quad E = \begin{bmatrix} [0,0,1] & [0,1,0] & [1,0,0] \\ [0,1,0] & [0,0,1] & [0,0,1] \\ [1,0,0] & [0,0,1] & [0,0,1] \end{bmatrix}, \quad A = \begin{bmatrix} 0 & 1 & 1 \\ 1 & 0 & 0 \\ 1 & 0 & 0 \end{bmatrix}, \quad \boldsymbol{p} = \begin{bmatrix} 1/3 \\ 1/3 \\ 1/3 \end{bmatrix}$$

where $F(i) \in \mathbb{R}^2$ represents the color type of the node $i$, $E(i,j) \in \mathbb{R}^3$ denotes the edge type between nodes $i$ and $j$, $A$ represents the shortest path matrix of the graph and $\boldsymbol{p}$ represents the uniform weights on the nodes. Graph (c) in Figure 1 is another example where edge features are vector-valued, giving $\Omega = \mathbb{R}^2$, where the first dimension indicates the length of the edge and the second dimension indicates its width.

Now, relying on the Definition 2.1, we introduce the Fused Network Gromov-Wasserstein (FNGW) distance dedicated to the comparison of node and edge featured graphs [1].

**Definition 2.3** (FNGW Distance, Discrete Case). Given $g = (F, A, E, \boldsymbol{p})$ of size $m$, $\tilde{g} = (\tilde{F}, \tilde{A}, \tilde{E}, \tilde{\boldsymbol{p}})$ of size $\tilde{m}$ corresponding to two tuples of $\mathcal{G}$, and trade-off parameters $(\alpha, \beta) \in [0,1]^2$, the Fused Network Gromov-Wasserstein distance between them for $(p, q) \in [1, \infty]$ is written as :

$$\text{FNGW}_{\alpha,\beta,q,p}(g, \tilde{g}) = \min_{\pi \in \Pi(\boldsymbol{p}, \tilde{\boldsymbol{p}})} \mathcal{E}_{\alpha,\beta,q,p}((F, A, E), (\tilde{F}, \tilde{A}, \tilde{E}), \pi) \tag{1}$$

---

[1] A very recent paper (Kawano et al., 2024) introduced a similar notion under the name of Multidimensional Fused Gromov-Wasserstein discrepancy.

with

$$\mathcal{E}_{\alpha,\beta,q,p}((F,A,E),(\tilde{F},\tilde{A},\tilde{E}),\pi) = \left( \sum_{i,j,k,l} \left[ \alpha d_\Omega \left( E(i,k),\tilde{E}(j,l) \right)^q + \beta |A(i,k) - \tilde{A}(j,l)|^q \right.\right.$$
$$\left.\left. + (1-\alpha-\beta)d_\Psi \left( F(i),\tilde{F}(j) \right)^q \right]^p \pi_{k,l}\pi_{i,j} \right)^{\frac{1}{p}} \tag{2}$$

**Example 2.4** (FNGW Distance). By using the graph representation procedure described in Example 2.2, the FNGW distance between the graph (a) and the graph (b) illustrated in Figure 1 is 0.296 when $\alpha = \frac{1}{3}$, $\beta = \frac{1}{3}$, $p = 1$ and $q = 2$, while the FGW distance between them is 0.

**Computing the FNGW Distance:** Let us first examine the computational complexity of our distance in this discrete case. We define the 4-dimensional tensors $J(A,\tilde{A})$ and $L(E,\tilde{E})$ as follows:

$$J_{i,j,k,l}(A,\tilde{A}) = |A(i,k) - \tilde{A}(j,l)|^q \quad L_{i,j,k,l}(E,\tilde{E}) = d_\Omega \left( E(i,k),\tilde{E}(j,l) \right)^q \tag{3}$$

and the cost matrix $M(F,\tilde{F})$:

$$M_{i,j}(F,\tilde{F}) = d_\Psi \left( F(i),\tilde{F}(j) \right)^q \tag{4}$$

Choosing $p = 1$, we can rewrite

$$\mathcal{E}_{\alpha,\beta,q,p} = \langle (1-\alpha-\beta)M(F,\tilde{F}) + \beta J(A,\tilde{A}) \otimes \pi + \alpha L(E,\tilde{E}) \otimes \pi, \pi \rangle \tag{5}$$

Here, $\otimes$ operator gives a matrix of the form $(J \otimes \pi)_{i,j} = \sum_{k,l} J_{i,j,k,l}\pi_{k,l}$. Following Peyré et al. (2016), the term $J(A,\tilde{A}) \otimes \pi$ can be computed efficiently when $q = 2$. The computation of $L(E,\tilde{E}) \otimes \pi$ is similarly non-trivial, requiring $O(m^2\tilde{m}^2T)$ operations, where $T$ is the number of operations necessary to compute the distance $d_\Omega(E(i,k),\tilde{E}(j,l))$. However, by choosing again an appropriate metric space $(\Omega, d_\Omega)$ and $q$, its complexity can be reduced via the application of the tensor-matrix multiplication trick from Proposition 1 of Peyré et al. (2016).

**Proposition 2.5.** *When $\Omega = \mathbb{R}^T$ with its associated metric $d_\Omega(a,b) = \|a - b\|_{\mathbb{R}^T}$ and $q = 2$, the term $L(E,\tilde{E}) \otimes \pi$ becomes*

$$L(E,\tilde{E}) \otimes \pi = g(E)\boldsymbol{p}\mathbb{1}_{\tilde{m}}^\mathsf{T} + \mathbb{1}_m \tilde{\boldsymbol{p}}^\mathsf{T} h(\tilde{E})^\mathsf{T} - 2\sum_{t=1}^{T} E[t]\pi\tilde{E}[t]^\mathsf{T} \tag{6}$$

*where $g : \mathbb{R}^{m \times m \times T} \to \mathbb{R}^{m \times m}$ is expressed as $g(E)_{i,j} = \|E(i,j)\|_{\mathbb{R}^T}^2$, $h : \mathbb{R}^{\tilde{m} \times \tilde{m} \times T} \to \mathbb{R}^{\tilde{m} \times \tilde{m}}$ is expressed as by $h(\tilde{E})_{i,j} = \|\tilde{E}(i,j)\|_{\mathbb{R}^T}^2$ and the matrix $E[t](i,j) = E(i,j,t)$ for any $i,j,t$. It can hence be computed with the complexity $O(m^2\tilde{m}T + m\tilde{m}^2T)$.*

Hence, with a focus on computational efficiency, **we have elected to operate within the metric space $\Omega$ defined in Proposition 2.5, with $p = 1, q = 2$, for the applications to graphs targeted in this work.** In practice, calculating the FNGW distance amounts to solving a non-convex constrained quadratic optimization problem. Following Vayer et al. (2019), we use the Conditional Gradient Descent (CGD) (i.e, the Frank-Wolfe algorithm). The complete algorithm is given in Algorithm 1, with the gradient with respect to the transport plan $\pi^{(k-1)}$ having the following form:

$$G = (1-\alpha-\beta)M(F,\tilde{F}) + 2\beta J(A,\tilde{A}) \otimes \pi^{(k-1)} + 2\alpha L(E,\tilde{E}) \otimes \pi^{(k-1)} \tag{7}$$

**General form:** We now introduce the general definition of the FNGW distance, which extends both the NGW-distance (Chowdhury & Mémoli, 2019) and the FGW-distance (Vayer et al., 2020). We propose experiments illustrating this generalization in Appendix D.3 and Appendix D.4.

**Definition 2.6** (FNGW Distance, General Form). Let $\mathcal{G}$ be the set of tuples of the form $(X, \psi_X, \varphi_X, \omega_X, \mu_X)$ where $X$ is a polish space, $\psi_X : X \to \Psi$ is a bounded continuous measurable function from $X$ to a metric space $(\Psi, d_\Psi)$, $\varphi_X : X \times X \to \mathbb{R}$ is a bounded continuous measurable function, $\omega_X : X \times X \to \Omega$ is a

---

**Algorithm 1** Computation of the FNGW Distance by CGD

---

**input:** $g = (F, A, E, \boldsymbol{p})$, $\tilde{g} = (\tilde{F}, \tilde{A}, \tilde{E}, \tilde{\boldsymbol{p}})$ and trade-off parameters $(\alpha, \beta)$
**init:** $\pi^{(0)} = \boldsymbol{p}\tilde{\boldsymbol{p}}^{\mathsf{T}} \in \mathbb{R}^{m \times \tilde{m}}$
**for** $k = 1, \ldots, K$ **do**
    Calculate gradient: $G = \nabla_{\pi^{(k-1)}} \mathcal{E}_{\alpha,\beta}((F, A, E), (\tilde{F}, \tilde{A}, \tilde{E}), \pi^{(k-1)})$
    Solve the optimization problem with an OT solver: $\tilde{\pi}^{(k-1)} \in \arg\min_{\tilde{\pi} \in \Pi(\boldsymbol{p}, \tilde{\boldsymbol{p}})} \langle G, \tilde{\pi} \rangle$
    Update the optimal plan: $\pi^{(k)} = (1 - \gamma^{(k)})\pi^{(k-1)} + \gamma^{(k)}\tilde{\pi}^{(k-1)}$ with $\gamma^{(k)} \in (0, 1)$ given by line-search algorithm (See details in Appendix B).
**end for**
Calculate the distance: $\mathrm{FNGW}_{\alpha,\beta}(g, \tilde{g}) = \mathcal{E}_{\alpha,\beta}((F, A, E), (\tilde{F}, \tilde{A}, \tilde{E}), \pi^{(K)})$
**output:** $\mathrm{FNGW}_{\alpha,\beta}(g, \tilde{g})$ and $\pi^{(K)}$

---

bounded continuous measurable function from $X^2$ to a metric space $(\Omega, d_\Omega)$ and $\mu_X$ is a fully supported Borel probability measure. Given two tuples $g_X = (X, \psi_X, \varphi_X, \omega_X, \mu_X)$, $g_Y = (Y, \psi_Y, \varphi_Y, \omega_Y, \mu_Y)$ from $\mathcal{G}$ and trade-off parameters $(\alpha, \beta) \in [0, 1]^2$, the Fused Network Gromov-Wasserstein Distance between $g_X$ and $g_Y$ is defined for any $(p, q) \in [1, \infty]$ as follows:

$$\mathrm{FNGW}_{\alpha,\beta,q,p}(g_X, g_Y) = \min_{\mu \in \Pi(\mu_X, \mu_Y)} \mathcal{E}_{\alpha,\beta,q,p}(g_X, g_Y, \mu) \tag{8}$$

with

$$\mathcal{E}_{\alpha,\beta,q,p}(g_X, g_Y, \mu) = \left( \int_{X \times Y} \int_{X \times Y} [(1 - \alpha - \beta)d_\Psi (\psi_X(x), \psi_Y(y))^q \right.$$

$$\left. + \alpha d_\Omega(\omega_X(x, x'), \omega_Y(y, y'))^q + \beta|\varphi_X(x, x') - \varphi_Y(y, y')|^q]^p d\mu(x, y)d\mu(x', y') \right)^{\frac{1}{p}} \tag{9}$$

The particular form of the FNGW distance in Def. 2.3 aligns with the scenario in which we assume that $X$ is a finite set of size $m$ and $\mu_X = \sum_i^m \boldsymbol{p}_i \delta_{x_i}$. As a consequence, $F$ is the result of evaluation of $\psi_X$ on every point of $X$, while $A$ and $E$ are outcomes of evaluating $\varphi_X$ and $\omega_X$ on every pair of points in $X^2$. Note that $\varphi$ are not necessarily symmetric functions, which corresponds to the setting of the NGW (Chowdhury & Mémoli, 2019), as opposed to the GW distance. We do not impose that constraint on the newly introduced functions $\omega$ either. In the context of edge-featured graphs, $\psi_X$ serves as the node labeling function, $\varphi_X$ represents the function that calculates the shortest path between two nodes, and $\omega_X$ acts as the edge labeling function.

The following theorem states the existence of the optimal coupling, so that the FNGW distance is well defined.

**Theorem 2.7** (Optimal Coupling). *Given* $g_X = (X, \psi_X, \varphi_X, \omega_X, \mu_X)$, $g_Y = (Y, \psi_Y, \varphi_Y, \omega_Y, \mu_Y)$, *for any* $(p, q) \in [1, \infty]$ *and* $(\alpha, \beta) \in [0, 1]^2$, *there exists an optimal coupling* $\mu^* \in \Pi(\mu_X, \mu_Y)$ *which satisfies* $\mathrm{FNGW}_{\alpha,\beta,q,p}(g_X, g_Y) = \mathcal{E}_{\alpha,\beta,q,p}(g_X, g_Y, \mu^*)$.

The following theorem states the metric properties of FNGW, which allows its application to a wide range of machine learning algorithms that only require pairwise comparisons.

**Theorem 2.8** (Metric Properties). *The FNGW distance satisfies the following properties: for all* $g_X = (X, \psi_X, \varphi_X, \omega_X, \mu_X)$, $g_Y = (Y, \psi_Y, \varphi_Y, \omega_Y, \mu_Y)$ *and* $g_Z = (Z, \psi_Z, \varphi_Z, \omega_Z, \mu_Z)$ *from* $\mathcal{G}$:

- *(Positivity)* $\mathrm{FNGW}_{\alpha,\beta,q,p}(g_X, g_Y) \geq 0$

- *(Symmetry)* $\mathrm{FNGW}_{\alpha,\beta,q,p}(g_X, g_Y) = \mathrm{FNGW}_{\alpha,\beta,q,p}(g_Y, g_X)$

- *(Equality)* $\mathrm{FNGW}_{\alpha,\beta,q,p}(g_X, g_X) = 0$. *Moreover,* $\mathrm{FNGW}_{\alpha,\beta,q,p}(g_X, g_Y) = 0$ *if and only there is a Borel probability space* $(Z, \mu_Z)$ *with measurable maps* $f : Z \to X$ *and* $g : Z \to Y$ *such that*

$$f_\#\mu_Z = \mu_X \quad g_\#\mu_Z = \mu_Y \tag{10}$$

$$\|(1 - \alpha - \beta)d_\Psi (\psi_X \circ f, \psi_Y \circ g)^q + \alpha d_\Omega(f^\# \omega_X, g^\# \omega_Y)^q + \beta|f^\# \varphi_X - g^\# \varphi_Y|^q\|_\infty = 0 \tag{11}$$

where $f^{\#}\omega_X : Z \times Z \to \Omega$ is the pullback weight function defined by $(z, z') \to \omega_X(f(z), f(z'))$, $f^{\#}\varphi_X : Z \times Z \to \mathbb{R}$ is given by $(z, z') \to \varphi_X(f(z), f(z'))$, and $g^{\#}$ is defined similarly.

- *(Relaxed Triangle Inequality)* $\mathrm{FNGW}_{\alpha,\beta,q,p}(g_X, g_Z) \leq 2^{q-1}(\mathrm{FNGW}_{\alpha,\beta,q,p}(g_X, g_Y) + \mathrm{FNGW}_{\alpha,\beta,q,p}(g_Y, g_Z))$

Equations (10)-(11) define a weak isomorphism between the objects of $\mathcal{G}$, while the last property states a relaxed triangle inequality with a factor of $2^{q-1}$. Consequently, when $q = 1$, FNGW is a proper metric over $\mathcal{G}$ endowed with such a weak isomorphism as an equivalence relation.

It should be noted that although FNGW theoretically satisfies the metric properties defined above, its practical guarantees depend on the specific computation algorithm used.

## 2.2 Fused Network Gromov-Wasserstein Barycenter and Computation

The notion of (weighted) barycenter can be encountered in many approaches in data science. Considering graph data, some recent works have successfully exploited GW-based barycenters in graph clustering (Peyré et al., 2016; Vayer et al., 2019) or in supervised graph prediction (Brogat-Motte et al., 2022). Similarly, our FNGW-based barycenter can be directly applied to the clustering of labeled graphs; we provide an example in Appendix D.4. Additionally, Graph Dictionary Learning (GDL) is feasible with the FNGW barycenter, where each graph is represented as the barycenter of the graphs in the dictionary. We develop this algorithm in Appendix B.2 and present an example in Appendix D.5. In what follows, we give a formal definition of such a barycenter, based on our proposed FNGW distance, and describe an algorithm to compute it.

**Definition 2.9** (FNGW Barycenter). Given a set $\{g_i\}_{i=1}^n$ with $g_i = (F_i, A_i, E_i, \boldsymbol{p}_i) \in \mathbb{R}^{m_i \times S} \times \mathbb{R}^{m_i \times m_i} \times \mathbb{R}^{m_i \times m_i \times T} \times \Sigma_{m_i}$ and a set of weights $\{\lambda_i\}_{i=1}^n$ such that $\sum_i \lambda_i = 1$, the FNGW Barycenter for a pre-defined histogram $\boldsymbol{p} \in \Sigma_m$ is defined as follows:

$$\mathfrak{B}(\{\lambda_i\}_i, \{g_i\}_i, \boldsymbol{p}) = \underset{F \in \mathbb{R}^{m \times S}, A \in \mathbb{R}^{m \times m}, E \in \mathbb{R}^{m \times m \times T}}{\arg\min} \sum_i \lambda_i \mathrm{FNGW}_{\alpha,\beta}((F, A, E, \boldsymbol{p}), g_i) \tag{12}$$

The above optimization problem can be reformulated as:

$$\min_{A, E, F, (\pi_i \in \Pi(\boldsymbol{p}, \boldsymbol{p}_i))_i} \sum_i \lambda_i \mathcal{E}_{\alpha,\beta} ((F, A, E), (F_i, A_i, E_i), \pi_i) \tag{13}$$

To obtain the FNGW barycenter, we employ the Block Coordinate Descent (BCD) algorithm, which means that we carry out the minimization in Equation 13 iteratively with respect to $\{\pi_i\}_i$, $F$, $A$ and $E$. The minimization with respect to $E$ has a closed form, which is given in the following proposition:

**Proposition 2.10.** *Optimizing Equation 13 with respect to tensor $E$ has a closed-form solution:*

$$E = \frac{1}{\mathcal{I}_{m \times T} \times_2 \boldsymbol{p}\boldsymbol{p}^{\mathsf{T}}} \sum_i \lambda_i (E_i \times_2 \pi_i) \times_1 \pi_i \tag{14}$$

The complete optimization algorithm is detailed in Algorithm 2.

We can notice that the optimization procedure preserves an interesting property for the tensor $E$ of the resulting barycenter:

**Proposition 2.11.** *If the set of tensors $\{E_i\}_i$ satisfies the condition: $\forall j, l, i, \sum_t^T E_i(j, l, t) = a \in \mathbb{R}$, then the barycenter $E$ given by Algorithm 2 also verify the same property.*

**Remark 2.12.** When the set of edge features for the graphs $\{g_i\}_{i=1}^n$ lies in a simplex space, the above proposition gives us the guarantee that the edge features of their barycenter will also be a simplex. For example, when the edge labels of the graphs are represented using one-hot encoding, the resulting barycenter can be discretized into a true graph by applying a simple *argmax* operation on the edge features, due to their simplex nature. This can be seen as a generalization of the *thresholding* operation on the adjacency matrix.

---

**Algorithm 2** Computation of FNGW Barycenter with BCD

---

**input:** $\{g_i\}_i$, fixed histogram $\boldsymbol{p}$, trade-off parameter $(\alpha, \beta)$
**init:** Randomly initialize $E^{(0)}, F^{(0)}$ and $A^{(0)}$.
**for** $k = 1, \ldots, K$ **do**
    Calculate $\{\pi_i\}_i$ with Alg. 1: $\pi_i^{(k)} = \arg\min_{\pi_i \in \Pi(\boldsymbol{p}, \boldsymbol{p_i})} \mathcal{E}_{\alpha, \beta}\left((F^{(k-1)}, A^{(k-1)}, E^{(k-1)}), (F_i, A_i, E_i), \pi_i\right)$
    Update $E$: $E^{(k)} = \frac{1}{\mathcal{I}_{m \times T \times 2}\boldsymbol{pp}^{\mathsf{T}}} \sum_i \lambda_i (E_i \times_2 \pi_i^{(k)}) \times_1 \pi_i^{(k)}$        $\triangleright$ Proposition 2.10
    Update $A$: $A^{(k)} = \frac{1}{\boldsymbol{pp}^{\mathsf{T}}} \sum_i \lambda^i \pi_i^{(k)} A_i \pi_i^{(k)\mathsf{T}}$        $\triangleright$ Proposition 4 in Peyré et al. (2016)
    Update $F$: $F^{(k)} = \sum_i \lambda_i \mathrm{diag}(\frac{1}{\boldsymbol{p}}) \pi_i^{(k)} F_i$        $\triangleright$ Equation 8 in Cuturi & Doucet (2014)
**end for**
**output:** The barycenter $(F^{(K)}, A^{(K)}, E^{(K)})$

---

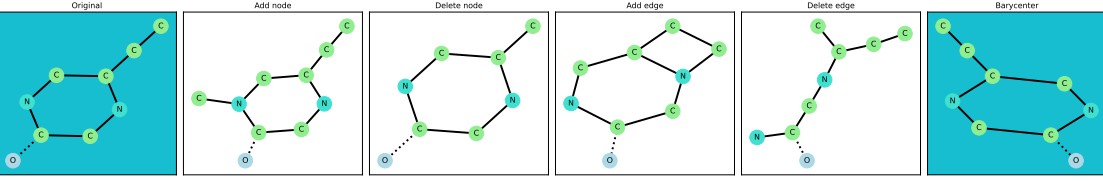

Figure 2: FNGW barycenter (rightmost) of the graphs obtained by perturbing a random molecule (leftmost).

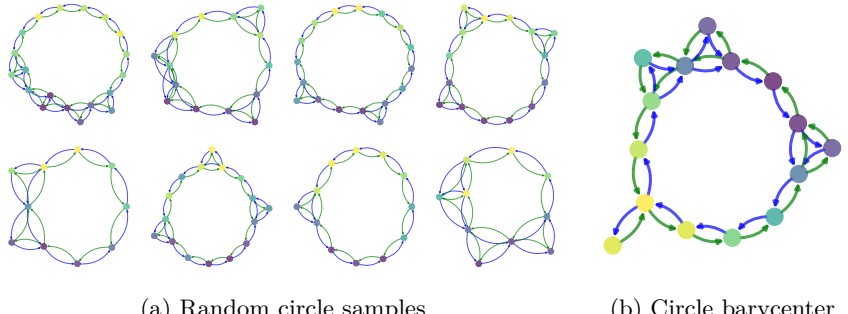

(a) Random circle samples        (b) Circle barycenter

Figure 3: Illustration of graph barycenters with the FNGW distance. The color of the node indicates the scalar value of the node feature.

The proofs of Proposition 2.10 and 2.11 can be found in Appendix A.

**Example 2.13** (Barycenter Computation)**.** To test the proposed barycenter computation algorithm, we perturb a random molecule using the following operations: node addition, node deletion, edge addition, and edge deletion. The resulting graphs are then used to compute the barycenter, successfully retrieving the original molecule before the perturbation, as shown in Figure 2. The FNGW distance between the barycenter (before discretization) and the original graph is 0.0079, with $\alpha = \frac{1}{3}$, $\beta = \frac{1}{3}$, $p = 1$, and $q = 2$. Another example of barycenter computation is shown in Figure 3. We randomly create 8 circle graphs, for which the number of nodes is randomly drawn between 10 and 20. Node features are scalars following a Sine function variation with additive Gaussian Noise $\mathcal{N}(0, 0.3)$. There exists two directed edges between each pair of adjacent nodes, the ascending one being colored in blue and the descending one in green. Supplementary edges are generated with probability 0.3 between every pair of nodes separated by another node. The toy graphs are therefore node-labeled, edge-labeled, and directed. The samples are shown in Figure 3a. To compute the FNGW barycenter, we take $A$ as the adjacency matrix, and encode the presence of colors in the one-hot tensor $E$. We choose the number of nodes of the barycenter to be 15, and the resulting graphs are shown in Figure 3b. We observe that both node and edge features are well preserved in the predicted barycenter, showing the effectiveness of Algorithm 2.

In the following section, we explore the application of the FNGW barycenter to supervised graph prediction.

## 3 Supervised Graph Prediction with the FNGW Barycenter

Supervised Graph Prediction task consists in learning to predict an output graph $g$ from a given input $x$ in input space $\mathcal{X}$. An appealing and original application of Optimal Transport for graphs consists in using a Gromov-Wasserstein distance as a loss in this supervised task (Brogat-Motte et al., 2022).

Let us define $\mathcal{G}$, the set of labeled graphs with maximum number of nodes $m_{\max}$.

$$\mathcal{G} = \big\{ (F, A, E, \boldsymbol{p}) \mid m_g \leq m_{\max},\ A \in \{0,1\}^{m_g \times m_g},\ F = (F_i)_{i=1}^{m_g} \in \mathcal{F}^{m_g}, $$
$$E = (E_{ij}) \in \mathcal{T}^{m_g \times m_g},\ \boldsymbol{p} = m_g{}^{-1} \mathbb{1}_{m_g} \big\} \tag{15}$$

where $\mathcal{F} \subset \mathbb{R}^S$ and $\mathcal{T} \subset \mathbb{R}^T$ are finite node feature and edge feature spaces, and $m_{\max}$ is the upper graph size. Denote $\mathcal{G}_m$ the relaxed version of set $\mathcal{G}$:

$$\mathcal{G}_m = \big\{ (F, A, E, \boldsymbol{p}) \mid A \in [0,1]^{m \times m},\ F = (F_i)_{i=1}^{m} \in \mathrm{Conv}(\mathcal{F})^{m}, $$
$$E = (E_{ij}) \in \mathrm{Conv}(\mathcal{T})^{m \times m},\ \boldsymbol{p} = m^{-1} \mathbb{1}_{m} \big\} \tag{16}$$

where $\mathrm{Conv}(\cdot)$ denotes the convex hull of the set.

We consider a set of training pairs consisting of inputs and graphs to be predicted $\{(x_i, g_i)\}_{i=1}^n$ drawn from a fixed but unknown distribution $\rho$ on $\mathcal{X} \times \mathcal{G}$. We are interested in the relaxed supervised graph prediction problem, i.e., finding a estimator $f : \mathcal{X} \to \mathcal{G}_m$ of the minimizer $f^*$ of the expected risk $\mathcal{R}(f) = \mathbb{E}_\rho[\mathrm{FNGW}_{\alpha,\beta}(f(X), G)]$ where the $\mathrm{FNGW}_{\alpha,\beta}$-distance's definition is extended to $\mathcal{G}_m \times \mathcal{G}$. We choose, as a solution to this problem, an estimator based on surrogate least square regression (Ciliberto et al., 2020) proposed by Brogat-Motte et al. (2022) that expresses as a barycenter of the output training data weighted by a function $\xi$ of the input $x$ :

$$\hat{f}(x) = \underset{g \in \mathcal{G}_m}{\arg\min} \sum_{i=1}^n \xi(x)_i \mathrm{FNGW}_{\alpha,\beta}(g, g_i) \tag{17}$$

with $\xi(x) = (\mathbf{K} + n\lambda I)^{-1} \boldsymbol{\kappa}_x$, where $\mathbf{K} \in \mathbb{R}^{n \times n}$ is the Gram matrix of the positive definite kernel $k$ defined on the input space $\mathcal{X}$ such that $\mathbf{K}_{ij} = k(x_i, x_j)$, $\boldsymbol{\kappa}_x = (k(x, x_1), \ldots, k(x, x_n))^{\mathsf{T}} \in \mathbb{R}^n$, and $\lambda > 0$ is a ridge regularization parameter. Note that for a given $x$, $\hat{g} \in \mathcal{G}$ is obtained by discretization of $\hat{f}(x)$.

The proposed estimator relies on the Implicit Loss Embedding (ILE) condition(Ciliberto et al., 2020), which expresses that an ILE loss can be written as an inner product of feature maps in some well-chosen Hilbert space. The ILE framework justifies the relevance of solving a surrogate regression problem in the so-called output feature space and ensures theoretical guarantees for this class of estimators. The next proposition guarantees that the estimator proposed in Equation 17 fits ILE condition.

**Proposition 3.1.** *The FNGW loss admits an Implicit Loss Embedding (ILE).*

Informally, this implies that our estimator is universally consistent and its learning rate is of order $n^{-1/4}$ with additional assumptions. The proof is given in Appendix A.6.

**Sketched ILE for big data regime.**  To avoid computational issues when scaling-up the number of samples, we propose a sketched version of the previous estimator. Sketching (Woodruff et al., 2014) applied on kernel approximation (Rudi et al., 2015; Avron et al., 2017; El Ahmad et al., 2023) leverages random projections of the Gram matrix, allowing to work with low-rank matrices and thus reducing compute time. Here, it gives $\xi(x) = \mathbf{K} S^{\mathsf{T}} (S \mathbf{K}^2 S^{\mathsf{T}} + n\lambda S \mathbf{K} S^{\mathsf{T}})^{\dagger} S \boldsymbol{\kappa}_x$ where $S \in \mathbb{R}^{s \times n}$ with $s \ll n$ is a sketching random matrix.

In order to conduct the prediction process described by Equation 17, we distinguish two possible situations. When a limited set of candidates from a set $\mathcal{G}_c$ is provided, one can resort to computing the score for each candidate. Otherwise, we resort to the method described in Alg. 2. We will showcase the two situations respectively with the experiments presented in Section 4.2 and Section 4.3.

## 4 Numerical Experiments

In this section, we conduct a series of studies around the proposed distance; then, we assess its relevance in supervised graph prediction problems, including the Fingerprint to Molecule task and Metabolite Identification task. A Python implementation of the algorithms presented for these tasks can be found on Github[2]. We leave additional experiments on clustering for Appendix D.

### 4.1 Study of the FNGW Distance

We begin our investigation by studying the potential benefit brought by our FNGW distance to graph-level supervised classification tasks. Then, we use these tasks to investigate how well the associated loss converges, and the impact of its hyperparameters on performance.

#### 4.1.1 Edge-featured Graph Classification

We represent input graphs to be classified by kernels based on the FNGW-distance; hence, we choose to use a variety of datasets where both node and edge features are present: Cuneiform (Kriege et al., 2018), MUTAG (Debnath et al., 1991; Kriege & Mutzel, 2012), PTC-MR (Helma et al., 2001; Kriege & Mutzel, 2012), BZR-MD, COX2-MD, DHFR-MD and ER-MD (Sutherland et al., 2003; Kriege & Mutzel, 2012). All these datasets were collected by Kersting et al. (2016). Our classifier is a Support Vector Machine (SVM) for which the kernel matrix $K$ is computed with $K_{i,j} = \exp(-\gamma \text{FNGW}(g_i, g_j))$. The kernel defined by FNGW is indefinite. While there exist in the literature SVM algorithms dedicated to indefinite kernels (see for instance (Ong et al., 2004; Luss & D' aspremont, 2007)), in these experiments, we rely on the classic SVM implementation. To compute the FNGW distance, we consider $F$ as the matrix of node features, and we set $A$ to be the matrix of shortest path distance and $E$ as the tensor of edge features where the non-edge position is attributed to a random vector of the same dimension. Details about the node and edge features present in each dataset, as well as details about graph representation for the FNGW computation are given in Appendix C.1.

We choose to use both kernel-based methods and graph neural networks (GNNs) for our baselines. We include a large set of kernels in our experiments: the Shortest Path Kernel (SPK) (Borgwardt & Kriegel, 2005), the Random Walk Kernel (RWK) (Gärtner et al., 2003), the Weisfeler Lehman Kernel (WLK) (Shervashidze et al., 2011), the Graphlet Sampling Kernel (GSK) (Shervashidze et al., 2009), the Neighborhood Subgraph Pairwise Distance Kernel (NSPDK) (Costa & Grave, 2010), the Hopper Kernel (HOPPERK) (Feragen et al., 2013), the Edge Histogram Kernel (EHK), the Vertex-edge Histogram Kernel (VEHK) (Sugiyama & Borgwardt, 2015) and the Propagation Kernel (PROPAK) (Neumann et al., 2016). For the GNNs, we consider the Principal Neighbourhood Aggregation (PNA) (Corso et al., 2020) and the Graph Attention Networks (GAT) (Veličković et al., 2018). Lastly, we also include in our experiments a kernel-based classifier induced by the FGW distance, computed with the same matrices $F$ and $A$. It should be noted that similarly to this last baseline, models don't necessarily use all available features.

We perform nested cross-validation with 50 iterations of the outer CV loop and report our graph classification results in Table 1. In Appendix C.1, we provide detailed information on hyperparameter search for all methods conducted during the inner CV loop. Our FNGW-based classifier outperforms consistently other graph kernel methods and GNN-based classifiers on most of the datasets. Furthermore, FNGW offers performances similar to FGW on PTC-MR, and significantly improves them on all the others datasets, except for MUTAG. We conjecture that the mutagenic effect on a bacterium (MUTAG) or carcinogenicity on rodents (PTC-MR) of molecules may not be sensitive to their chemical bond type, which is the information encoded in the edge features. But when combined with information about the distance between atoms, these features seem useful for distinguishing between active and inactive molecules (BZR-MD, COX2-MD, DHFR-MD, ER-MD).

---

[2]https://github.com/chunchiehy/fngw

Table 1: Graph classification performance on real datasets. The best results are highlighted in **Bold**. The last seven rows show the results for methods leveraging the edge features.

| Methods | Cuneiform | MUTAG | PTC-MR | BZR-MD | COX2-MD | DHFR-MD | ER-MD |
|---|---|---|---|---|---|---|---|
| RWK | $73.93 \pm 7.73$ | $88.42 \pm 7.59$ | $57.43 \pm 7.61$ | $71.81 \pm 8.48$ | $62.65 \pm 7.93$ | $67.20 \pm 5.67$ | $70.22 \pm 6.32$ |
| SPK | $74.00 \pm 7.39$ | $84.00 \pm 8.80$ | $57.37 \pm 8.16$ | $71.10 \pm 6.71$ | $64.90 \pm 8.43$ | $65.85 \pm 5.87$ | $69.78 \pm 6.14$ |
| GSK | $15.63 \pm 6.84$ | $83.89 \pm 8.04$ | $54.86 \pm 8.24$ | $48.13 \pm 6.73$ | $44.90 \pm 6.94$ | $67.45 \pm 6.33$ | $59.16 \pm 6.44$ |
| WLK-VH | $73.70 \pm 7.34$ | $88.00 \pm 7.95$ | $61.71 \pm 6.18$ | $63.10 \pm 6.92$ | $56.71 \pm 7.82$ | $66.65 \pm 5.54$ | $66.98 \pm 6.19$ |
| HOPPERK | $74.00 \pm 7.39$ | $83.89 \pm 9.50$ | $54.29 \pm 7.36$ | $73.81 \pm 7.00$ | $61.10 \pm 7.86$ | $67.60 \pm 5.85$ | $70.84 \pm 6.97$ |
| PROPAK | $73.26 \pm 7.49$ | $76.84 \pm 9.59$ | $58.57 \pm 8.09$ | $73.10 \pm 7.03$ | $58.58 \pm 8.45$ | $66.85 \pm 6.14$ | $66.44 \pm 6.34$ |
| FGW | $86.59 \pm 7.51$ | $86.11 \pm 8.34$ | $63.77 \pm 9.20$ | $73.61 \pm 6.60$ | $64.90 \pm 7.43$ | $62.70 \pm 5.95$ | $72.09 \pm 7.14$ |
| NSPDK | $77.48 \pm 8.91$ | $\mathbf{88.63 \pm 5.99}$ | $59.94 \pm 8.33$ | $73.10 \pm 7.03$ | $60.00 \pm 8.70$ | $67.30 \pm 6.46$ | $63.16 \pm 7.41$ |
| EHK | $5.63 \pm 3.65$ | $67.68 \pm 9.76$ | $55.43 \pm 7.73$ | $66.97 \pm 7.34$ | $60.26 \pm 7.11$ | $67.35 \pm 6.35$ | $70.09 \pm 7.20$ |
| VEHK | $74.00 \pm 7.39$ | $77.26 \pm 9.32$ | $58.97 \pm 8.14$ | $72.52 \pm 7.67$ | $\mathbf{80.06 \pm 7.09}$ | $66.80 \pm 7.65$ | $75.24 \pm 6.93$ |
| WLK-VEH | $73.48 \pm 7.46$ | $82.53 \pm 8.90$ | $62.23 \pm 7.07$ | $64.00 \pm 7.25$ | $58.65 \pm 8.48$ | $66.65 \pm 5.54$ | $71.20 \pm 5.83$ |
| PNA | $84.15 \pm 8.05$ | $83.79 \pm 8.01$ | $59.94 \pm 7.33$ | $71.14 \pm 9.60$ | $71.48 \pm 7.99$ | $67.60 \pm 6.30$ | $77.24 \pm 6.51$ |
| GAT | $37.19 \pm 11.90$ | $75.26 \pm 10.38$ | $57.94 \pm 7.94$ | $70.20 \pm 7.46$ | $70.13 \pm 9.93$ | $69.10 \pm 5.80$ | $72.40 \pm 6.89$ |
| FNGW | $\mathbf{89.41 \pm 6.06}$ | $83.05 \pm 8.95$ | $\mathbf{63.94 \pm 7.54}$ | $\mathbf{75.68 \pm 7.01}$ | $68.90 \pm 8.33$ | $\mathbf{73.45 \pm 7.28}$ | $\mathbf{79.78 \pm 5.85}$ |

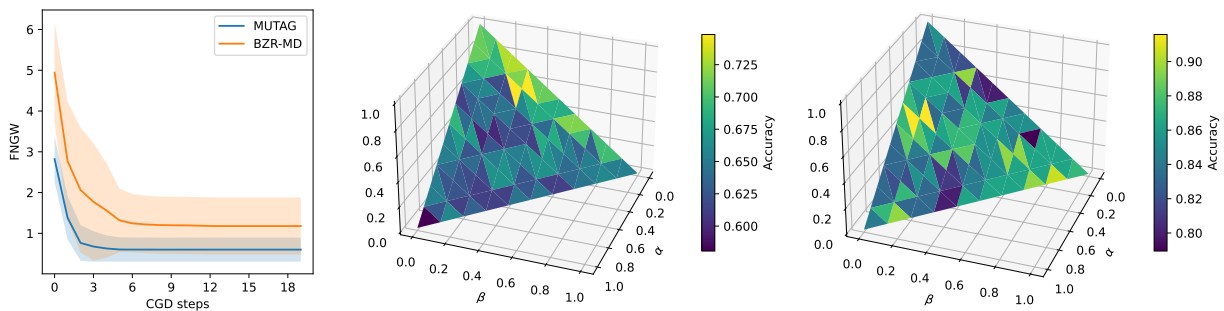

Figure 4: (Left) Convergence rate of FNGW with the CGD algorithm. (Middle) Classification accuracy on MUTAG w.r.t different values of $\alpha$ and $\beta$. (Right) Classification accuracy on BZR-MD w.r.t different values of $\alpha$ and $\beta$. The optimal values of $\alpha$ and $\beta$ vary across different datasets.

### 4.1.2 Convergence of the FNGW Distance

The left panel of Figure 4 shows the convergence of the FNGW distance using the conditional gradient descent (CGD) in Algorithm 1 on the MUTAG and BZR-MD datasets. Each computation step involves randomly selecting two graphs, with this procedure being repeated 10 times. We can see that during this computation, the FNGW converges in fewer than 10 steps of CGD, on both datasets.

### 4.1.3 Impact of the Parameters Presented in the FNGW

We explore the influence of parameters $\alpha$ and $\beta$ on the FNGW distance on classification performance. The heat maps illustrating test set accuracies for the MUTAG and BZR-MD datasets are presented respectively in the central and right panels of Figure 4. These visuals show that for $\alpha$, which governs the weighting of edge features, a small value (even approaching zero) yields commendable accuracy on MUTAG. Conversely, it is a larger $\alpha$ that seems to give optimal results for BZR-MD. These observations reinforce our previous hypothesis: it does not seem useful to exploit edge features on MUTAG, which could explain why FNGW lags behind FGW for that dataset.

## 4.2 Supervised Graph Prediction: Fingerprint to Molecule

In this experiment, we delve into to a novel graph prediction problem. The aim is to predict a molecule from its *fingerprint* representation: hence, we will use here our FNGW distance as a loss, through Supervised Graph Prediction estimators, as defined in Section 3. It should be noted that this is a very difficult end-to-end task as it implies to predict new molecules rather than selecting them from a candidate set.

**Dataset.** To conduct our experiment, we have chosen the widely used QM9 molecule dataset (Ruddigkeit et al., 2012; Ramakrishnan et al., 2014). QM9 is a collection of small organic molecules, totaling around 130,000 in number. We use a subset of 2,000 molecules as test set. The molecules are kekulized [3] by the chemical software RDKit [4], accompanied by the removal of hydrogen atoms. After the pre-processing, each molecule contains up to 9 atoms of Carbon, Nitrogen, Oxygen, or Fluorine. There is three types of edges: single, double, and triple bonds. We adopt a fingerprint representation of the molecules as input features; these fingerprints are generated using the Morgan algorithm provided by RDKit, with a specified radius of 2 and a size of 2048. For each molecule, we represent it by an atom matrix $F \in \mathbb{R}^{m \times 4}$ and a bond tensor $E \in \mathbb{R}^{m \times m \times 3}$ where $m$ is the number of the atoms. Both the atom features and bond features are encoded by one-hot encoding. To apply the FNGW distance on the molecules of QM9, we use the uniform measure over the atoms and the parameter $\beta$ is set to 0. We name the resulting dataset **Fin2Mol**.

**Graph Edit Distance and Correlation with FNGW.** We utilize Graph Edit Distance (GED) as a metric to assess the effectiveness of graph prediction, given its widespread use in measuring the similarity between pairs of graphs. Given two graphs $g_1$ and $g_2$, the graph edit distance between them is defined by:

$$\mathrm{GED}(g_1, g_2) = \min_{(o_1, \cdots, o_k) \in \mathcal{O}(g_1, g_2)} \sum_i^k c(o_i) \qquad (18)$$

where $\mathcal{O}(g_1, g_2)$ is the set of all edit paths allowing to transform $g_1$ into a graph isomorphic to $g_2$ and $c(o) \geq 0$ is the cost of each elementary edit operation $o$ which is one of vertex substitution, edge substitution, vertex deletion, edge deletion, vertex insertion, and edge insertion. We use the GED implementation provided by the package NetworkX (Hagberg et al., 2008) and the cost of each elementary edit operation is set to one.

In Figure 5, the left panel shows the correlation between GED and FNGW across various values of $\alpha$. Specifically, we randomly select 1000 pairs of molecules from the QM9 dataset and compute both the GED and the FNGW distance for each pair. Subsequently, we determine the Pearson and Spearman correlation coefficients between the GED and the FNGW distance. The results reveal a strong correlation between the FNGW and the GED, particularly for commonly chosen values of $\alpha$. This correlation demonstrates well the interest of employing FNGW as loss function for graph prediction: FNGW effectively serves as a viable proxy or approximation for GED, which is prohibitively expensive to compute in practice, and is not differentiable.

**Choice of the Parameter $\alpha$.** While $\alpha$, as an hyper-parameter of the FNGW loss, can be selected via cross-validation, our preceding observation prompts us to propose an alternative approach. Indeed, we can use correlation between the GED and the FNGW values resulting from $\alpha$ as signal to select the latter. Subsequently, the $\alpha$ value yielding the strongest correlation with the GED is selected to define the training loss in the graph prediction model. The underlying philosophy of this strategy is straightforward: by ensuring consistence between the training loss and the evaluation metric, we anticipate achieving a good performance.

**Experimental Settings.** We build five training-test splits using different seeds. Each split has 131,885 training samples and 2,000 test samples. We choose to compare our ILE-FNGW with ILE-FGW (Brogat-Motte et al., 2022). Additionally, we adapt the NNBary estimator proposed by Brogat-Motte et al. (2022) with FNGW loss to study its effect. NNBary expresses the predicted graph as the barycenter of several graph templates, which are jointly learned with the coefficients of the templates parameterized by a neural network. At prediction time, we provide the true number of atoms to the estimators in order to compute the barycenter. For the ILE estimator, we choose the kernel $k$ used to compute the coefficient $\xi(x)$ to be Gaussian; given the large quantity of data, we use SubSample sketching (Rudi et al., 2015) for the input kernel approximation. For the NNBary estimator, a MLP of one hidden layer is used to encode input features

---

[3]Kekulize: converts aromatic rings to their Kekule form. `https://www.rdkit.org/docs/RDKit_Book.html`
[4]RDKit: Open-source cheminformatics. `https://www.rdkit.org`

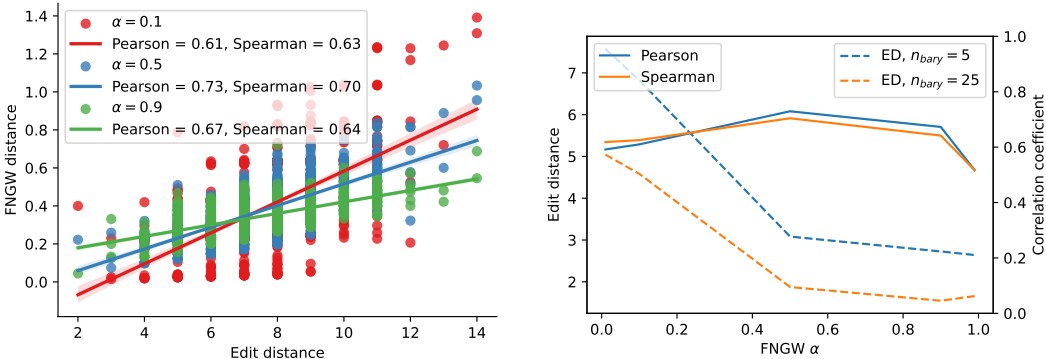

Figure 5: (Left) A strong correlation exsits between the FNGW and the GED for commonly chosen values of $\alpha$. (Right) The performances of ILE-FNGW and the correlation coefficients between GED and FNGW with various values of $\alpha$. The optimal $\alpha$ values for correlation generally lead to favorable performance.

Table 2: Graph edit distances of different methods on the Fin2Mol test set. The symbol ‡ indicates the result where $\alpha$ is chosen based on the correlation coefficient, while in other cases, it is selected through cross-validation. Best results are in **Bold**.

|  | GED w/o edge feature ↓ | GED w/ edge feature ↓ |
|---|---|---|
| NNBary-FGW | $5.000 \pm 0.140$ | - |
| NNBary-FNGW | $5.311 \pm 0.090$ | $5.756 \pm 0.073$ |
| Sketched ILE-FGW | $3.037 \pm 0.111$ | - |
| Sketched ILE-FNGW | $\mathbf{1.449 \pm 0.034}$ | $\mathbf{1.534 \pm 0.029}$ |
| Sketched ILE-FNGW ‡ | $1.739 \pm 0.068$ | $1.809 \pm 0.065$ |

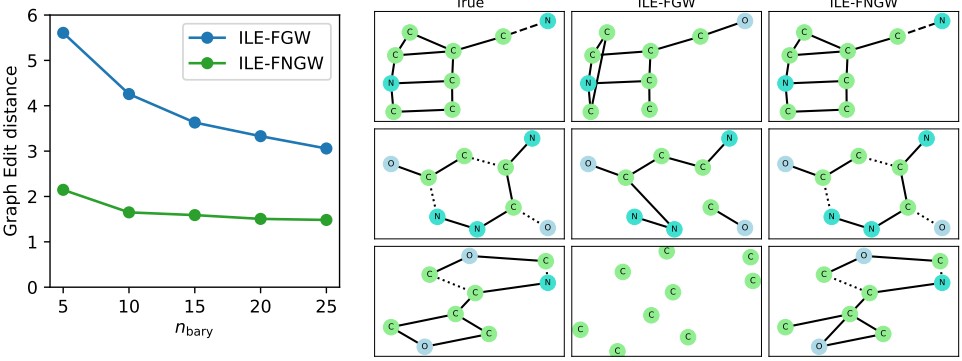

Figure 6: (Left) Quantitative results with respect the number of training points used to compute the prediction. Fewer training points are needed for the barycenter with ILE-FNGW while it provides better performance than ILE-FGW. (Right) Qualitative comparison of the predicted QM9 molecules.

into templates weights. The FNGW parameter $\alpha$, the ridge regularization parameter $\lambda$, the parameter of the Gaussian kernel $\gamma$, the sketching size, the number of points used to compute the barycenter and the dimension of the NNBary MLP are chosen through cross-validation over a subset of size 500. We provide an additional result on ILE with FNGW, where $\alpha$ is chosen based on the correlation coefficient with GED, as described above. More details about the cross-validation are given in Appendix C.2.

Table 3: Top-$k$ accuracies on the metabolite identification test set. Best results are in **Bold**.

|  | Top-1 ↑ | Top-10 ↑ | Top-20 ↑ |
|---|---|---|---|
| WL kernel | 9.8% | 29.1% | 37.4% |
| IOKR - Fingerprint w/ linear kernel | 28.6% | 54.5% | 59.9% |
| IOKR - Fingerprint w/ gaussian kernel | **41.0**% | **62.0**% | **67.8**% |
| ILE-FGW diffuse | 28.1% | 53.6% | 59.9% |
| ILE-FNGW diffuse + Bond stereo | 27.7% | 55.2% | 60.9% |
| ILE-FNGW diffuse + Bond type | 34.6% | 55.1% | 60.0% |
| ILE-FNGW diffuse + Mix | 36.2% | 58.2% | 61.9% |

**Experimental Results.** Our results are presented in Table 2. They clearly indicate that FNGW, as loss function, outperforms FGW with ILE and achieves comparable performance to FGW with NNBary. The results for NNBary can be attributed to the model's limited expressibility. For ILE, we envision two different explanations, which may overlap: firstly, from a learning standpoint, it is conceivable that FNGW induces a more meaningful implicit graph embedding than FGW, thereby facilitating the resolution of the surrogate regression problem. Secondly, from an optimization viewpoint, the incorporation of bond type constraints within FNGW aids in computing the barycenters, consequently resulting in more realistic molecules. Additionally, FNGW's ability to predict the bond types of molecules enables us to evaluate with a version of the GED that takes edge features into account. The final row of Table 2 highlights that selecting $\alpha$ using the best Pearson/Spearman correlation coefficient yields comparable performance to what is achieved through cross-validation. Additionally, as depicted in the right panel of Figure 5,the optimal $\alpha$ values for correlation generally lead to favorable performance of predictions. Finally, the left panel of Figure 6 shows the influence of varying number of training points used in the barycenter computation on the performance of ILE . It is evident that increasing the size of training data generally enhances model performance. However, beyond a threshold of 15 training points, the observed improvement becomes less significant. A visual comparison showcasing predicted graphs generated by various methods is presented in the right panel of Figure 6. The first two examples demonstrate how the FNGW method enhances predictions related to atom types and molecule structure, while the final example validates its efficacy concerning the optimizing during the barycenter computation. See Appendix D.6 for more examples of prediction.

### 4.3 Supervised Graph Prediction: Metabolite Identification

In this last experiment, our task is again to predict a molecule; however, this application differs from the first one, as a candidate set $\mathcal{G}_c$ is given at inference time.

**Dataset.** As in Brouard et al. (2016a); Brogat-Motte et al. (2022), we use the Metabolite Identification dataset processed by Dührkop et al. (2015). The learning algorithm is expected to predict the metabolites given a tandem mass spectra. For each spectra, a set of possible metabolites of the same chemical formula is provided. The candidate sets were built with molecular structures from PubChem. The dataset is split into a training set of size $n = 3000$ and a test set of size $n_{\text{test}} = 1022$. More details about the dataset can be found in Appendix C.3.

**Experimental Settings.** In order to represent the metabolites, we encode their adjacency into $A$ and the atom types as one-hot vectors into $F$. Following Brogat-Motte et al. (2022), we use the matrix diffused by the normalized Laplacian of the adjacency matrix: $F_{\text{diff}} = e^{-\tau \text{Lap}(A)} F$. To obtain $E$, we use three configurations: the chemical bond type, the chemical bond stereochemistry (which are embedded through one-hot encodings) and the concatenation of both (*Mix*). The input representation is obtained through a probability product kernel (Heinonen et al., 2012) on the input mass spectra, which has been shown effective on this problem (Brouard et al., 2016a). Due to the computational cost, during prediction, only the 5 training samples with the greatest weights $\xi(x)_i$ are taken into account, rather than all the samples, as described in Equation 17. Details about the hyperparameter choices are given in Appendix C.3.

**Experimental Results.** We measure the performance of various models on Metabolite Identification via Top-$k$ accuracy with $k \in \{1, 10, 20\}$. The results are presented in Table 3. Results for the WL kernel and IOKR(Brouard et al., 2016b) are taken from Brogat-Motte et al. (2022). We observe that with mixed edge features, FNGW significantly outperforms FGW. Our method approaches the performance of fingerprint with a Gaussian kernel, which uses an expert-derived molecular representation. This results further confirms that using a more complete representation of the output space, able to exploit more structure information, is crucial for graph prediction.

## 5 Conclusion

We propose a novel Optimal Transport distance for pairwise graph comparison in the presence of edge features, unlocking many applications where this information is available and relevant. This distance is built on the Network Gromov-Wasserstein and Fused Gromov-Wassertein distances and inherits similar geometric properties; to apply it on graph data, we devise algorithms to compute both the novel distance and the associated barycenter in the discrete case. Comprehensive experimentation on our proposed distance gives a deeper understanding of the behavior of FNGW as a tool for graph representation, and empirical evaluation on real-world datasets shows that the use of this distance as a loss function yields significant improvements in terms of performance in supervised graph prediction tasks. Future works will be dedicated to scale-up both the distance computation and the barycenter computation algorithms to target very large graph datasets. We also plan to investigate propagating edge information to nodes and reverse.

**Acknowledgments**

The authors are grateful to Luc Brogat-Motte for the insightful discussions at the beginning of the project. They also thank reviewers for their comments and suggestions. The first author is supported by a PhD grant provided by Hi! PARIS Center.

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

## A Technical Proofs

In Sections A.1 and A.2, we present the proofs of the theoretical properties of the FNGW distance that can appear as natural extensions of the properties of the NGW distance Chowdhury & Mémoli (2019) and the FGW distance Vayer et al. (2020). We therefore leverage results from both Chowdhury & Mémoli (2019) and Vayer et al. (2020), and additionally the metric properties of space $\Omega$ for our proofs of Theorem 2.7 and 2.8. In Sections A.3, A.4 and A.5, we provide proofs and calculation of the expressions we need when computing the distance itself and the barycenter. In Section A.6, we demonstrate the ILE property for the FNGW distance.

For the sake of completeness, we recall first the definitions of the NGW and the FGW distance.

**Definition A.1** (Network Gromov-Wasserstein Distance (Chowdhury & Mémoli, 2019))**.** Let $\mathcal{G}$ be the set of tuple of the form $(X, \varphi_X, \mu_X)$ where $X$ is a polish space, $\varphi_X : X \times X \to \mathbb{R}$ is **a bounded continuous measurable function** and $\mu_X$ is a fully supported Borel probability measure. Given two tuples $g_X = (X, \varphi_X, \mu_X)$, $g_Y = (Y, \varphi_Y, \mu_Y)$ from $\mathcal{G}$, the Network Gromov-Wasserstein Distance between $g_X$ and $g_Y$ is defined for any $p \in [1, \infty]$ as follows:

$$\mathrm{NGW}_p(g_X, g_Y) = \min_{\mu \in \Pi(\mu_X, \mu_Y)} \mathcal{E}_p(g_X, g_Y, \mu) \tag{19}$$

with

$$\mathcal{E}_p(g_X, g_Y, \mu) = \left( \int_{X \times Y} \int_{X \times Y} |\varphi_X(x, x') - \varphi_Y(y, y')|^p d\mu(x, y) d\mu(x', y') \right)^{\frac{1}{p}} \tag{20}$$

It should be noted that the FNGW distance extends NWG into a *fused* case and generalize the codomain space of $\varphi$ from $\mathbb{R}$ to a metric space. In the discrete case, when applied to graphs, we leverage this property to take into account edge labels as feature vectors.

**Definition A.2** (Fused Gromov-Wasserstein Distance (Vayer et al., 2020))**.** Let $\mathcal{G}$ be the set of tuple of the form $(X, \psi_X, \varphi_X, \mu_X)$ where $X$ is a polish space, $\psi_X : X \to \Psi$ is a bounded continuous measurable function from $X$ to a metric space $(\Psi, d_\Psi)$ , $\varphi_X : X \times X \to \mathbb{R}_+$ is a **metric**, and $\mu_X$ is a fully supported Borel probability measure. Given two tuples $g_X = (X, \psi_X, \varphi_X, \mu_X)$, $g_Y = (Y, \psi_Y, \varphi_Y, \mu_Y)$ from $\mathcal{G}$ and trade-off parameter $\alpha \in [0, 1]$, the Fused Network Gromov-Wasserstein Distance between $g_X$ and $g_Y$ is defined for any $(p, q) \in [1, \infty]$ as follows:

$$\mathrm{FGW}_{\alpha, q, p}(g_X, g_Y) = \min_{\mu \in \Pi(\mu_X, \mu_Y)} \mathcal{E}_{\alpha, q, p}(g_X, g_Y, \mu) \tag{21}$$

with

$$\mathcal{E}_{\alpha, q, p}(g_X, g_Y, \mu) = \left( \int_{X \times Y} \int_{X \times Y} [(1 - \alpha) d_\Psi \left( \psi_X(x), \psi_Y(y) \right)^q \right.$$
$$\left. + \alpha |\varphi_X(x, x') - \varphi_Y(y, y')|^q]^p d\mu(x, y) d\mu(x', y') \right)^{\frac{1}{p}} \tag{22}$$

Compared with the FGW distance, FNGW fuses another more general function: $\omega_X : X \times X \to \Omega$, while releasing the symmetric constraint of $\varphi$ and allowing for feature vectors as edge labels.

### A.1 Proof of Theorem 2.7: Existence of FNGW Distance

The proof takes mainly advantage of the Weierstrass theorem, which will use the following lemmas:

**Lemma A.3** (Compactness of Couplings; Lemma 10, Chowdhury & Mémoli (2019)). *Let $X$, $Y$ be two Polish spaces and let $\mu_X \in \mathrm{Prob}(X), \mu_Y \in \mathrm{Prob}(Y)$. Then $\Pi(\mu_X, \mu_Y)$ is compact in $\mathrm{Prob}(X \times Y)$.*

**Lemma A.4** (Continuity of the Functional $\mu \mapsto \mathcal{E}_{\alpha,\beta,q,p}(g_X, g_Y, \mu)$). *For $(p, q) \in [1, \infty]^2$, let $g_X = (X, \psi_X, \varphi_X, \omega_X, \mu_X)$, $g_Y = (Y, \psi_Y, \varphi_Y, \omega_Y, \mu_Y)$ both from $\mathcal{G}$, then the functional*

$$\Pi(\mu_X, \mu_Y) \to \mathbb{R}_+ \bigcup +\infty$$
$$\mu \mapsto \mathcal{E}_{\alpha,\beta,q,p}(g_X, g_Y, \mu)$$

*is lower semicontinuous on $\Pi(\mu_X, \mu_Y)$ for the weak convergence of measures.*

*Proof.* We define the functional $f : X \times Y \times X \times Y \to \mathbb{R}_+ \bigcup +\infty$ with

$$f((x,y),(x',y')) = [(1 - \alpha - \beta)d_\Psi\left(\psi_X(x), \psi_Y(y)\right)^q + \alpha d_\Omega(\omega_X(x,x'), \omega_Y(y,y'))^q$$
$$+ \beta|\varphi_X(x,x') - \varphi_Y(y,y')|^q]^p \tag{23}$$

then, $f$ is lower semicontinuous due to the continuity of $(d_\Psi, d_\Omega)$ and $(\psi_X, \psi_Y, \omega_X, \omega_Y, \varphi_X, \varphi_Y)$. Using Lemma 3 from Vayer et al. (2020) by considering $W = X \times Y$, which is a polish space, we can conclude $\mu \mapsto \mathcal{E}_{\alpha,\beta,q,p}(g_X, g_Y, \mu)$ is lower semicontinuous on $\Pi(\mu_X, \mu_Y)$ for the weak convergence of measures. $\square$

We can now prove Theorem 2.7, which mainly takes advantage of the Weierstrass theorem.

*Proof.* Since $\Pi(\mu_X, \mu_Y) \subset \mathrm{Prob}(X \times Y)$ is compact and the functional $\mu \to \mathcal{E}_{\alpha,\beta,q,p}(g_X, g_Y, \mu)$ is lower semicontinous on $\Pi(\mu_X, \mu_Y)$, we can conclude the functional achieves its infimum for some $\mu^*$ by applying directly the Weierstrass theorem. $\square$

### A.2 Proof of Theorem 2.8: Metric Properties of FNGW Distance.

We divide Theorem 2.8 into four lemmas, and we suppose $g_X = (X, \psi_X, \varphi_X, \omega_X, \mu_X)$, $g_Y = (Y, \psi_Y, \varphi_Y, \omega_Y, \mu_Y)$ and $g_Z = (Z, \psi_Z, \varphi_Z, \omega_Z, \mu_Z)$ are from $\mathcal{G}$.

**Lemma A.5** (Positivity). $\mathrm{FNGW}_{\alpha,\beta,q,p}(g_X, g_Y) \geq 0$

*Proof.* $\mathrm{FNGW}_{\alpha,\beta,q,p}(g_X, g_Y) \geq 0$ since $d_\Psi$ and $d_\Omega$ are both metrics and $(\alpha, \beta) \in [0, 1]^2$. $\square$

**Lemma A.6** (Symmetry). $\mathrm{FNGW}_{\alpha,\beta,q,p}(g_X, g_Y) = \mathrm{FNGW}_{\alpha,\beta,q,p}(g_Y, g_X)$

*Proof.* For any $\mu \in \Pi(\mu_X, \mu_Y)$, let $\mu^\star := T_\# \mu$ be the push forward of $\mu$ via a Borel map $T$ defined as follows

$$T : X \times Y \to Y \times X$$
$$(x, y) \mapsto (y, x)$$

Then we have, (by the property of the push forward)

$$\mathcal{E}_{\alpha,\beta,q,p}(g_Y, g_X, \mu^\star)$$

$$=\left(\int_{(Y \times X)^2} \left[(1 - \alpha - \beta)d_\Psi(\psi_Y(y), (\psi_X(x))^q + \alpha d_\Omega(\omega_Y(y, y'), \omega_X(x, x'))^q\right.\right.$$

$$\left.\left. + \beta|\varphi_Y(y, y') - \varphi_X(x, x')|^q\right]^p d\mu^\star(y, x)d\mu^\star(y', x')\right)^{\frac{1}{p}}$$

$$=\left(\int_{(X \times Y)^2} \left[(1 - \alpha - \beta)d_\Psi(\psi_Y(y), (\psi_X(x))^q + \alpha d_\Omega(\omega_Y(y, y'), \omega_X(x, x'))^q\right.\right.$$

$$\left.\left. + \beta|\varphi_Y(y, y') - \varphi_X(x, x')|^q\right]^p d\mu(x, y)d\mu(x', y')\right)^{\frac{1}{p}}$$

$$=\left(\int_{(X \times Y)^2} \left[(1 - \alpha - \beta)d_\Psi(\psi_X(x), \psi_Y(y))^q + \alpha d_\Omega(\omega_X(x, x'), \omega_Y(y, y'))^q\right.\right.$$

$$\left.\left. + \beta|\varphi_X(x, x') - \varphi_Y(y, y')|^q\right]^p d\mu(x, y)d\mu(x', y')\right)^{\frac{1}{p}}$$

$$=\mathcal{E}_{\alpha,\beta,q,p}(g_X, g_Y, \mu)$$

The first equality is given by property of the push forward ($\int f dT_{\#\mu} = \int f \circ T d\mu$); the second equality is given by the symmetry of $d_\Psi$ and $d_\Omega$. As a consequence, we have $\text{FNGW}_{\alpha,\beta,q,p}(g_Y, g_X) = \text{FNGW}_{\alpha,\beta,q,p}(g_X, g_Y)$. $\square$

**Lemma A.7** (Equality). $\text{FNGW}_{\alpha,\beta,q,p}(g_X, g_X) = 0$. *Moreover,* $\text{FNGW}_{\alpha,\beta,q,p}(g_X, g_Y) = 0$ *if and only there is a Borel probability space* $(Z, \mu_Z)$ *with measurable maps* $f : Z \to X$ *and* $g : Z \to Y$ *such that*

$$f_{\#}\mu_Z = \mu_X \tag{24}$$

$$g_{\#}\mu_Z = \mu_Y \tag{25}$$

$$\|(1 - \alpha - \beta)d_\Psi(\psi_X \circ f, \psi_Y \circ g)^q + \alpha d_\Omega(f^{\#}\omega_X, g^{\#}\omega_Y)^q + \beta|f^{\#}\varphi_X - g^{\#}\varphi_Y|^q\|_\infty = 0 \tag{26}$$

*where* $f^{\#}\omega_X : Z \times Z \to \Omega$ *is the pullback weight function defined by* $(z, z') \to \omega_X(f(z), f(z'))$, $f^{\#}\varphi_X : Z \times Z \to \mathbb{R}$ *is given by* $(z, z') \to \varphi_X(f(z), f(z'))$, *and* $g^{\#}$ *is defined similarly.*

*Proof.* The proof is analogous to the proof of Theorem 18 in Chowdhury & Mémoli (2019). We first deal with the case where $p \in [1, +\infty)$. For the backward direction, let us assume that there exists a Borel probability space $(Z, \mu_Z)$ and measurable maps $f : Z \to X$, $g : Z \to Y$ verifying Equation 24 - 26. We consider $\mu^\star := (f, g)_{\#}\mu_Z$. First of all, it is easy to prove that $\mu^\star \in \mathscr{C}(\mu_X, \mu_Y)$:

$$\forall A \in \mathscr{B}(X), \mu^\star(A \times Y) = \mu_Z((f, g)^{-1}[A \times Y]) = \mu_Z(f^{-1}[A]) = \mu_X(A)$$

$$\forall B \in \mathscr{B}(Y), \mu^\star(X \times B) = \mu_Z((f, g)^{-1}[X \times B]) = \mu_Z(g^{-1}[B]) = \mu_Y(B)$$

Then we have

$$\text{FNGW}_{\alpha,\beta,q,p}(g_X, g_Y)$$

$$\leq\left(\int_{(X \times Y)^2} \left[(1 - \alpha - \beta)d_\Psi(\psi_X(x), \psi_Y(y))^q + \alpha d_\Omega(\omega_X(x, x'), \omega_Y(y, y'))^q\right.\right.$$

$$\left.\left. + \beta|\varphi_X(x, x') - \varphi_Y(y, y')|^q\right]^p d\mu^\star(x, y)d\mu^\star(x', y')\right)^{\frac{1}{p}}$$

$$=\|(1 - \alpha - \beta)d_\Psi(\psi_X \circ f, \psi_Y \circ g)^q + \alpha d_\Omega(f^{\#}\omega_X, g^{\#}\omega_Y)^q + \beta|f^{\#}\varphi_X - g^{\#}\varphi_Y|^q\|_{L^p(\mu_Z \otimes \mu_Z)}$$

which is equal to 0. Conversely, let $\mu^\star \in \mathscr{C}(\mu_X, \mu_Y)$ be the optimal coupling satisfying $\text{FNGW}_{\alpha,\beta,q,p}(g_X, g_Y) = 0$. So we have $\|(1-\alpha-\beta)d_\Psi(\psi_X, \psi_Y)^q + \alpha d_\Omega(\omega_X, \omega_Y)^q + \beta|\varphi_X - \varphi_Y|^q\|_{L^p(\mu^\star \otimes \mu^\star)} = 0$. To prove the existence of the desired probability space and measurable maps, we define

$$Z := X \times Y, \mu_Z := \mu^\star$$

$$f := proj_X, g := proj_Y$$

where $proj_X : Z \to X$ and $proj_Y : Z \to Y$ are projection maps. It can be shown that

$$\forall A \in \mathscr{B}(X), f_{\#}\mu_Z(A) = \mu^{\star}(f^{-1}[A]) = \mu^{\star}(A \times Y) = \mu_X(A)$$
$$\forall B \in \mathscr{B}(Y), g_{\#}\mu_Z(B) = \mu^{\star}(g^{-1}[B]) = \mu^{\star}(X \times B) = \mu_Y(B)$$

and

$$\|(1 - \alpha - \beta)d_\Psi(\psi_X \circ f, \psi_Y \circ g)^q + \alpha d_\Omega(f^{\#}\omega_X, g^{\#}\omega_Y)^q + \beta|f^{\#}\varphi_X - g^{\#}\varphi_Y|^q\|_\infty$$
$$= \|(1 - \alpha - \beta)d_\Psi(\psi_X, \psi_Y)^q + \alpha d_\Omega(\omega_X, \omega_Y)^q + \beta|\varphi_X - \varphi_Y|^q\|_\infty = 0$$

The proof for the case $p = \infty$ is analogous.

For the specific case $g_X = g_Y$, we consider $(Z, \mu_Z) = (X, \mu_X)$ and identity maps $(f, g)$, then Equation 24 - 26 are well verified, so we have $\mathrm{FNGW}_{\alpha,\beta,q,p}(g_X, g_X) = 0$. The proof is thus concluded. $\qquad\square$

**Lemma A.8** (Relaxed Triangle Inequality)**.**

$$\mathrm{FNGW}_{\alpha,\beta,q,p}(g_X, g_Z) \leq 2^{q-1}(\mathrm{FNGW}_{\alpha,\beta,q,p}(g_X, g_Y) + \mathrm{FNGW}_{\alpha,\beta,q,p}(g_Y, g_Z))$$

*Proof.* Let $\mu_{ab}$ be the optimal coupling between $(g_X, g_Y)$ and $\mu_{bc}$ be the optimal coupling between $(g_Y, g_Z)$. By the Gluing Lemma, there exists a probability mesure $\tilde{\mu} \in \mathrm{Prob}(X, Y, Z)$ with marginals $\mu_{ab}$ on $(X \times Y)$ and $\mu_{bc}$ on $(Y \times Z)$. Let $\mu_{ac}$ be the marginal of $\tilde{\mu}$ on $(X \times Z)$. Due to the fact that $\mu_{ac}$ is not necessary a optimal coupling between $(g_X, g_Z)$, we have

$$
\begin{aligned}
&\mathrm{FNGW}_{\alpha,\beta,q,p}(g_X, g_Z) \\
\leq &\|(1 - \alpha - \beta)d_\Psi(\psi_X, \psi_Z))^q + \alpha d_\Omega(\omega_X, \omega_Z)^q + \beta|\varphi_X - \varphi_Z|^q\|_{L^p(\mu_{ac} \otimes \mu_{ac})} \\
\leq &\|(1 - \alpha - \beta)(d_\Psi(\psi_X, \psi_Y) + d_\Psi(\psi_Y, \psi_Z))^q + \alpha(d_\Omega(\omega_X, \omega_Y) + d_\Omega(\omega_Y, \omega_Z))^q \\
&+ \beta(|\varphi_X - \varphi_Y| + |\varphi_Y - \varphi_Z|)^q\|_{L^p(\tilde{\mu} \otimes \tilde{\mu})} \\
\leq &\|(1 - \alpha - \beta)2^{q-1}(d_\Psi(\psi_X, \psi_Y)^q + d_\Psi(\psi_Y, \psi_Z)^q) + \alpha 2^{q-1}(d_\Omega(\omega_X, \omega_Y)^q + d_\Omega(\omega_Y, \omega_Z)^q) \\
&+ \beta 2^{q-1}(|\varphi_X - \varphi_Y|^q + |\varphi_Y - \varphi_Z|^q)\|_{L^p(\tilde{\mu} \otimes \tilde{\mu})} \\
= &2^{q-1}\|((1 - \alpha - \beta)d_\Psi(\psi_X, \psi_Y)^q + \alpha d_\Omega(\omega_X, \omega_Y)^q + \beta|\varphi_X - \varphi_Y|^q) \\
&+ ((1 - \alpha - \beta)d_\Psi(\psi_Y, \psi_Z)^q + \alpha d_\Omega(\omega_Y, \omega_Z)^q + \beta|\varphi_Y - \varphi_Z|^q)\|_{L^p(\tilde{\mu} \otimes \tilde{\mu})} \\
\leq &2^{q-1}(\|(1 - \alpha - \beta)d_\Psi(\psi_X, \psi_Y)^q + \alpha d_\Omega(\omega_X, \omega_Y)^q + \beta|\varphi_X - \varphi_Y|^q\|_{L^p(\tilde{\mu} \otimes \tilde{\mu})} \\
&+ \|(1 - \alpha - \beta)d_\Psi(\psi_Y, \psi_Z)^q + \alpha d_\Omega(\omega_Y, \omega_Z)^q + \beta|\varphi_Y - \varphi_Z|^q\|_{L^p(\tilde{\mu} \otimes \tilde{\mu})}) \\
= &2^{q-1}(\|(1 - \alpha - \beta)d_\Psi(\psi_X, \psi_Y)^q + \alpha d_\Omega(\omega_X, \omega_Y)^q + \beta|\varphi_X - \varphi_Y|^q\|_{L^p(\mu_{ab} \otimes \mu_{ab})} \\
&+ \|(1 - \alpha - \beta)d_\Psi(\psi_Y, \psi_Z)^q + \alpha d_\Omega(\omega_Y, \omega_Z)^q + \beta|\varphi_Y - \varphi_Z|^q\|_{L^p(\mu_{ab} \otimes \mu_{ab})}) \\
= &2^{q-1}(\mathrm{FNGW}_{\alpha,\beta,q,p}(g_X, g_Y) + \mathrm{FNGW}_{\alpha,\beta,q,p}(g_Y, g_Z))
\end{aligned}
$$

The second inequality is the result of the triangle inequality of the inner metrics, the third inequality is due to the fact that $\forall q \geq 1$, $\forall a, b \geq 0$, $(a + b)^q \leq 2^{q-1}(a^q + b^q)$. The fourth inequality is the consequence of the Minkowski's inequality of the norm $L^p(\tilde{\mu} \otimes \tilde{\mu})$. Finally, it proves the relaxed triangle inequality with a factor of $2^{q-1}$. When $q = 1$, the triangle inequality is well satisfied. $\qquad\square$

### A.3 Proof of Proposition 2.5: Computation Complexity Reduction for FNGW

*Proof.* Given tensor $E$, we define the matrix $E[t]$ by $E[t](i,k) = E(i,k,t)$ for any $i,k,t$. By the definition of tensor-matrix multiplication $\otimes$, we have

$$
\begin{aligned}
\left(L(E, \tilde{E}) \otimes \pi\right)_{i,j} &= \sum_{k,l} \|E(i,k) - \tilde{E}(j,l)\|_{\mathbb{R}^T}^2 \pi_{k,l} \\
&= \sum_{k,l} \sum_t |E(i,k,t) - \tilde{E}(j,l,t)|^2 \pi_{k,l} \\
&= \sum_t \sum_{k,l} \left(E(i,k,t)^2 + \tilde{E}(j,l,t)^2 - 2E(i,k,t)\tilde{E}(j,l,t)\right) \pi_{k,l} \\
&= \sum_t \sum_{k,l} \left(E[t](i,k)^2 + \tilde{E}[t](j,l)^2 - 2E[t](i,k)\tilde{E}[t](j,l)\right) \pi_{k,l}
\end{aligned}
\tag{27}
$$

We note that the inner sum over $k$ and $l$, in the last equation above, is the same as the one that is computed in the GW distance, considering that $E[t]$ and $\tilde{E}[t]$ are the similarity matrices. Taking advantage of Prop.1 in Peyré et al. (2016), the above equation becomes:

$$
\begin{aligned}
\left(L(E, \tilde{E}) \otimes \pi\right)_{i,j} &= \left(\sum_t E[t]^2 \boldsymbol{p} \mathbb{1}_{\tilde{m}}^\mathsf{T} + \mathbb{1}_n \tilde{\boldsymbol{p}}^\mathsf{T} \tilde{E}[t]^{2\mathsf{T}} - 2E[t]\pi\tilde{E}[t]^\mathsf{T}\right)_{i,j} \\
&= \left(g(E)\boldsymbol{p}\mathbb{1}_{\tilde{m}}^\mathsf{T} + \mathbb{1}_m \tilde{\boldsymbol{p}}^\mathsf{T} h(\tilde{E})^\mathsf{T} - 2\sum_{t=1}^T E[t]\pi\tilde{E}[t]^\mathsf{T}\right)_{i,j}
\end{aligned}
$$

where $g : \mathbb{R}^{m \times m \times T} \to \mathbb{R}^{m \times m}$ is defined by $g(E)_{i,j} = \|E(i,j)\|_{\mathbb{R}^T}^2$ and $h : \mathbb{R}^{\tilde{m} \times \tilde{m} \times T} \to \mathbb{R}^{\tilde{m} \times \tilde{m}}$ is defined by $h(\tilde{E})_{i,j} = \|\tilde{E}(i,j)\|_{\mathbb{R}^T}^2$. $\qquad\square$

### A.4 Proof of Proposition 2.10: Justification of the Barycenter Algorithm

*Proof.* Using Equation 5 and Equation 6, we can write

$$
\begin{aligned}
&\underset{E \in \mathbb{R}^{m \times m \times T}}{\arg\min} \sum_i \lambda_i \mathcal{E}_{\alpha,\beta}\left((F, A, E), (F_i, A_i, E_i), \pi_i\right) \\
&= \underset{E \in \mathbb{R}^{m \times m \times T}}{\arg\min} \sum_i \lambda_i \left\langle \sum_t E[t]^2 \boldsymbol{p}\mathbb{1}_{m_i}^\mathsf{T} + \mathbb{1}_m \boldsymbol{p_i}^\mathsf{T} E_i[t]^{2\mathsf{T}} - 2E[t]\pi_i E_i[t]^\mathsf{T}, \pi_i \right\rangle \\
&= \underset{E \in \mathbb{R}^{m \times m \times T}}{\arg\min} \sum_t \sum_i \lambda_i \left\langle E[t]^2 \boldsymbol{p}\mathbb{1}_{m_i}^\mathsf{T} + \mathbb{1}_m \boldsymbol{p_i}^\mathsf{T} E_i[t]^{2\mathsf{T}} - 2E[t]\pi_i E_i[t]^\mathsf{T}, \pi_i \right\rangle
\end{aligned}
$$

Now let us write the first-order optimality condition. If $E^*$ is a minimum of the previous expression, we have:

$$
\left(\nabla_E(\sum_t \sum_i \lambda_i \left\langle E[t]^2 \boldsymbol{p}\mathbb{1}_{m_i}^\mathsf{T} + \mathbb{1}_m \boldsymbol{p_i}^\mathsf{T} E_k[t]^{2\mathsf{T}} - 2E[t]\pi_i E_i[t]^\mathsf{T}, \pi_i \right\rangle\right)_{|E=E^*} = \boldsymbol{0}
$$

which reads

$$
\forall t, \sum_i \lambda_i \left(\nabla_{E[t]} \left\langle E[t]^2 \boldsymbol{p}\mathbb{1}_{m_i}^\mathsf{T} + \mathbb{1}_m \boldsymbol{p_i}^\mathsf{T} E_i[t]^{2\mathsf{T}} - 2E[t]\pi_i E_i[t]^\mathsf{T}, \pi_i \right\rangle\right)_{|E[t]=E^*[t]} = \boldsymbol{0}
$$

We notice that for each $t$ we have the same optimization problem as the one described by Equation 12 of Peyré et al. (2016). Taking advantage of Proposition 13 of Peyré et al. (2016) and with the notations of the tensor operations, we obtain the desired solution. $\qquad\square$

## A.5 Proof of Proposition 2.11: Property of the FNGW Barycenter

*Proof.* By Equation14, we have the expression for each element of the barycenter tensor (we omit here the iteration index for the sake of clarity):

$$E(s,r,t) = \frac{1}{\boldsymbol{p}_s \boldsymbol{p}_r} \sum_i \lambda_i \sum_j \sum_l \pi_i(s,j) E_i(j,l,t) \pi_i^\mathsf{T}(l,r)$$

Summing it up along the third dimension, we have

$$
\begin{aligned}
\sum_t E(s,r,t) &= \sum_t \frac{1}{\boldsymbol{p}_s \boldsymbol{p}_r} \sum_i \lambda_i \sum_j \pi_i(s,j) \sum_l E_i(j,l,t) \pi_i^\mathsf{T}(l,r) \\
&= \frac{1}{\boldsymbol{p}_s \boldsymbol{p}_r} \sum_i \lambda_i \sum_j \pi_i(s,j) \sum_l \pi_i(r,l) \sum_t E_i(j,l,t) \\
&= a \frac{1}{\boldsymbol{p}_s \boldsymbol{p}_r} \sum_i \lambda_i \sum_j \pi_i(s,j) \sum_l \pi_i(r,l) \\
&= a \frac{1}{\boldsymbol{p}_s \boldsymbol{p}_r} \sum_i \lambda_i \boldsymbol{p}_s \boldsymbol{p}_r \\
&= a
\end{aligned}
$$

$\square$

## A.6 Proof of Proposition 3.1: Statistical Guarantees for Supervised Graph Prediction Estimator

Before going through the proof, let us recall the definition of ILE property.

**Definition A.9** (ILE, Ciliberto et al. (2020))**.** A continuous map $\ell \colon \mathcal{Z} \times \mathcal{Y} \to \mathbb{R}$ is said to admit an Implicit Loss Embedding (ILE) if there exists a separable Hilbert Space $\mathcal{H}$ and two measurable bounded maps $\psi : \mathcal{Z} \to \mathcal{H}$ and $\varphi : \mathcal{Y} \to \mathcal{H}$, such that for any $z \in \mathcal{Z}$ and $y \in \mathcal{Y}$ we have

$$\ell(z,y) = \langle \psi(z), \varphi(y) \rangle_\mathcal{H}, \tag{28}$$

and $\|\varphi(y)\|_\mathcal{H} \leq 1$.

Then we recall the $\mathrm{FNGW}_{\alpha,\beta}$-distance's definition when extended to $\mathcal{G}_m \times \mathcal{G}$. Given $g_m = \{F, E, A, \boldsymbol{p}\} \in \mathcal{G}_m$ and $g = \{\tilde{F}, \tilde{E}, \tilde{A}, \tilde{\boldsymbol{p}}\} \in \mathcal{G}$,

$$
\begin{aligned}
\mathrm{FNGW}_{\alpha,\beta}(g_m, g) = \min_{\pi \in \Pi(\boldsymbol{p}, \tilde{\boldsymbol{p}})} \sum_{i,j,k,l} \Big[ &\alpha \|E(i,k) - \tilde{E}(j,l)\|^2_{\mathbb{R}^T} + \beta |A(i,k) - \tilde{A}(j,l)|^2 \\
&+ (1 - \alpha - \beta) \|F(i) - \tilde{F}(j)\|^2_{\mathbb{R}^S} \Big] \pi_{k,l} \pi_{i,j}
\end{aligned}
\tag{29}
$$

*Proof.* Using Theorem 12 from (Ciliberto et al., 2020), we are going to show that $\mathrm{FNGW}_{\alpha,\beta}$ satisfies the ILE property by proving that i) $\mathcal{G}_m$ is compact, ii) $\mathcal{G}$ is finite (trivial with the definition) and iii) the function $\mathrm{FNGW}_{\alpha,\beta}(\cdot, g)$ is continuous.
First of all, we can see that $\mathcal{G}_m$ is compact, since $[0,1]^{m \times m}$, $\mathrm{Conv}(\mathcal{F})^m$, and $\mathrm{Conv}(\mathcal{T})^{m \times m}$ are compact ($\mathcal{F}$ and $\mathcal{T}$ are finite and thus compact). Secondly, $\mathcal{G}$ is finite by definition (Equation 15). Now, we will prove the continuity of $\mathrm{FNGW}_{\alpha,\beta}(\cdot, g)$ for any $g \in \mathcal{G}$. Denote $dg_m = (\mathrm{d}F, \mathrm{d}A, \mathrm{d}E) \in \mathcal{G}_m$. For a given $g \in \mathcal{G}$, we

have, for any $(g_m, dg_m)$:

$$|\text{FNGW}_{\alpha,\beta}(g_m + dg_m, g) - \text{FNGW}_{\alpha,\beta}(g_m, g)|$$

$$\leq \sup_{\pi \in \Pi(\boldsymbol{p}, \tilde{\boldsymbol{p}})} \sum_{i,j,k,l} \left[ \alpha \left( 2\langle dE(i,k), E(i,k) - \tilde{E}(j,l) \rangle_{\mathbb{R}^T} + \|dE(i,k)\|^2_{\mathbb{R}^T} \right) \right.$$

$$+ \beta \left( 2dA(i,k) \times (A(i,k) - \tilde{A}(j,l)) + dA(i,k)^2 \right)$$

$$\left. + (1 - \alpha - \beta) \left( 2\langle dF(i), F(i) - \tilde{F}(j) \rangle_{\mathbb{R}^S} + \|dF(i)\|^2_{\mathbb{R}^S} \right) \right] \pi_{k,l} \pi_{i,j}$$

$$\leq \sum_{i,j,k,l} \left[ \alpha \left( 2\|dE(i,k)\|_{\mathbb{R}^T} \|E(i,k) - \tilde{E}(j,l)\|_{\mathbb{R}^T} + \|dE(i,k)\|^2_{\mathbb{R}^T} \right) \right.$$

$$+ \beta \left( 2|dA(i,k)| \times |A(i,k) - \tilde{A}(j,l)| + dA(i,k)^2 \right)$$

$$\left. + (1 - \alpha - \beta) \left( 2\|dF(i)\|_{\mathbb{R}^S} \|F(i) - \tilde{F}(j)\|_{\mathbb{R}^S} + \|dF(i)\|^2_{\mathbb{R}^S} \right) \right] \tag{30}$$

The first inequality is due to the fact that $\forall f, g : \Pi(\boldsymbol{p}, \tilde{\boldsymbol{p}}) \to \mathbb{R}, |\min_\pi f(\pi) - \min_\pi g(\pi)| \leq \sup_\pi |f(\pi) - g(\pi)|$. The second is a direct application of the Cauchy-Schwarz inequality and the fact that $\forall i, j, \pi_{i,j} \leq 1$. Since $E$, $F$ and $A$ are all bounded, there exists $M \in \mathbb{R}$ such that for any $i, j, k, l$:

$$M \geq \|E(i,k) - \tilde{E}(j,l)\|_{\mathbb{R}^T} \tag{31}$$

$$M \geq |A(i,k) - \tilde{A}(j,l)| \tag{32}$$

$$M \geq \|F(i) - \tilde{F}(j)\|_{\mathbb{R}^S} \tag{33}$$

Then we have

$$|\text{FNGW}_{\alpha,\beta}(g_m + dg_m, g) - \text{FNGW}_{\alpha,\beta}(g_m, g)|$$

$$\leq \sum_{i,j,k,l} \left[ \alpha \left( 2\|dE(i,k)\|_{\mathbb{R}^T} M + \|dE(i,k)\|^2_{\mathbb{R}^T} \right) + \beta \left( 2|dA(i,k)| M + dA(i,k)^2 \right) \right.$$

$$\left. + (1 - \alpha - \beta) \left( 2\|dF(i)\|_{\mathbb{R}^S} M + \|dF(i)\|^2_{\mathbb{R}^S} \right) \right] \tag{34}$$

Now when $\|dg_m\|_{\mathbb{R}^{m \times m} \times \mathbb{R}^{m \times S} \times \mathbb{R}^{m \times m \times T}} \to 0$, we have $\|dF\|_{\mathbb{R}^{m \times S}} \to 0$, $\|dA\|_{\mathbb{R}^{m \times m}} \to 0$, and $\|dE\|_{\mathbb{R}^{m \times m \times T}} \to 0$, we can easily deduce from Equation 34 that

$$|\text{FNGW}_{\alpha,\beta}(g_m + dg_m, g) - \text{FNGW}_{\alpha,\beta}(g_m, g)| \underset{\|dg_m\| \to 0}{\longrightarrow} 0 \tag{35}$$

Hence, we have that for any $g \in \mathcal{G}$, $\text{FNGW}_{\alpha,\beta}(\cdot, g)$ is continuous on $\mathbb{R}^{m \times m} \times \mathbb{R}^{m \times S} \times \mathbb{R}^{m \times m \times T}$, and thus on $\mathcal{G}_m$. $\qquad \square$

Since $\text{FNGW}_{\alpha,\beta}$ admits an ILE, Theorems 8 and 9 from Ciliberto et al. (2020) can be instantiated on the $\text{FNGW}_{\alpha,\beta}$-based estimator in Equation 17 which gives us the following results.

**Theorem A.10** (Universal Consistency). *Let $k$ be a bounded universal reproducing kernel. For any $n \in \mathbb{N}$ and any distribution $\rho$ on $\mathcal{X} \times \mathcal{G}$, let $f_n : \mathcal{X} \to \mathcal{G}_m$ be the estimator defined in Equation 17 trained on $\{(x_i, y_i)\}^n_{i=1}$ samples independently drawn from $\rho$ and with $\lambda = n^{-1/2}$, then*

$$\lim_{n \to \infty} \mathcal{R}(f_n) = \mathcal{R}(f^*) \quad \text{with probability 1} \tag{36}$$

*with $\mathcal{R}(f) = \mathbb{E}_\rho[\text{FNGW}_{\alpha,\beta}(f(X), G)]$ and $f^*$ denotes the bayes estimator.*

**Theorem A.11** (Excess-risk Bounds). *Let $\mathcal{H}$ be the Hilbert space associated with the loss $\text{FNGW}_{\alpha,\beta} : \mathcal{G}_m \times \mathcal{G} \to \mathbb{R}_+$ in the ILE definition. Let $k : \mathcal{X} \times \mathcal{X} \to \mathbb{R}$ be a continuous reproducing kernel on $\mathcal{X}$ with associated RKHS $\mathcal{F}$ such that $\kappa^2 := \sup_{x \in \mathcal{X}} k(x, x) < +\infty$. Let $\rho$ be a distribution on $\mathcal{X} \times \mathcal{G}$ and suppose that the solution $h^*$ of the surrogate regression problem belongs to the considered hypothesis space $\mathcal{H} \otimes \mathcal{F}$.*

*Let $\delta \in (0,1]$ and $n_0$ sufficiently large such that $n_0^{-1/2} \geq \frac{9\kappa^2}{n_0} \log \frac{n_0}{\delta}$. Then, for any $n \geq n_0$, the estimator $f_n$ defined in Equation 17 trained on $n$ points independently sampled from $\rho$ and with $\lambda = n^{-1/2}$ is such that, with probability at least $1 - \delta$*

$$\mathcal{R}(f_n) - \mathcal{R}(f^*) \leq c \log(4/\delta) \; n^{-1/4}, \tag{37}$$

*with $c$ a constant independent of $n$ and $\delta$.*

### A.7 Proof of Invariance of FNGW to Node Permutation

**Proposition A.12** (Invariance of FNGW to Node Permutation). *Given graphs $g_1 = (F_1, A_1, E_1, \boldsymbol{p}_1)$, $g_2 = (F_2, A_2, E_2, \boldsymbol{p}_2)$, and a new graph $\hat{g}_1 = (\hat{F}_1, \hat{A}_1, \hat{E}_1, \hat{\boldsymbol{p}}_1)$ obtained by applying a node permutation $\sigma$ on $g_1$, then we have*

$$\mathrm{FNGW}_{\alpha,\beta,q,p}(g_1, g_2) = \mathrm{FNGW}_{\alpha,\beta,q,p}(\hat{g}_1, g_2) \tag{38}$$

*Proof.* Given two graphs $g_1 = (F_1, A_1, E_1, \boldsymbol{p}_1)$ and $g_2 = (F_2, A_2, E_2, \boldsymbol{p}_2)$, the FNGW distance between them is given by

$$\min_{\pi \in \Pi(\boldsymbol{p}_1, \boldsymbol{p}_2)} \left( \sum_{i=1}^{m_1} \sum_{k=1}^{m_1} \sum_{j,l} \left[ \alpha d_\Omega \left(E_1(i,k), E_2(j,l)\right)^q + \beta |A_1(i,k) - A_2(j,l)|^q + (1-\alpha-\beta) d_\Psi \left(F_1(i), F_2(j)\right)^q \right]^p \pi_{k,l} \pi_{i,j} \right)^{\frac{1}{p}} \tag{39}$$

Suppose that $\hat{g}_1 = (\hat{F}_1, \hat{A}_1, \hat{E}_1, \hat{\boldsymbol{p}}_1)$ is isomorphic to $g_1$ with a permutation application $\sigma : [\![1, m_1]\!] \to [\![1, m_1]\!]$, so the FNGW distance between the new permuted graph $\hat{g}_1$ and $g_2$ is

$$\min_{\hat{\pi} \in \Pi(\hat{\boldsymbol{p}}_1, \boldsymbol{p}_2)} \left( \sum_{i=1}^{m_1} \sum_{k=1}^{m_1} \sum_{j,l} \left[ \alpha d_\Omega \left(\hat{E}_1(i,k), E_2(j,l)\right)^q + \beta |\hat{A}_1(i,k) - A_2(j,l)|^q + (1-\alpha-\beta) d_\Psi \left(\hat{F}_1(i), F_2(j)\right)^q \right]^p \hat{\pi}_{k,l} \hat{\pi}_{i,j} \right)^{\frac{1}{p}} \tag{40}$$

Since $\hat{E}_1(i,k) = E_1(\sigma^{-1}(i), \sigma^{-1}(k))$, $\hat{A}_1(i,k) = A_1(\sigma^{-1}(i), \sigma^{-1}(k))$, $\hat{F}_1(i) = F_1(\sigma^{-1}(i))$, $\hat{\pi}_{k,l} = \pi_{\sigma^{-1}(k),l}$ and $\hat{\pi}_{i,j} = \pi_{\sigma^{-1}(i),j}$ due to the permutation, Equation 40 can be rewritten by

$$\min_{\pi \in \Pi(\boldsymbol{p}_1, \boldsymbol{p}_2)} \left( \sum_{i=1}^{m_1} \sum_{k=1}^{m_1} \sum_{j,l} \left[ \alpha d_\Omega \left(E_1(\sigma^{-1}(i), \sigma^{-1}(k)), E_2(j,l)\right)^q + \beta |A_1(\sigma^{-1}(i), \sigma^{-1}(k)) - A_2(j,l)|^q \right. \right.$$

$$\left. \left. + (1-\alpha-\beta) d_\Psi \left(F_1(\sigma^{-1}(i)), F_2(j)\right)^q \right]^p \pi_{\sigma^{-1}(k),l} \pi_{\sigma^{-1}(i),j} \right)^{\frac{1}{p}} \tag{41}$$

Denoting $s = \sigma^{-1}(i)$ and $t = \sigma^{-1}(k)$ and set $\mathcal{S} = \{\sigma^{-1}(i) \mid i \in [\![1, m_1]\!]\}$, we obtain thus

$$\min_{\pi \in \Pi(\boldsymbol{p}_1, \boldsymbol{p}_2)} \left( \sum_{s \in \mathcal{S}} \sum_{t \in \mathcal{S}} \sum_{j,l} \left[ \alpha d_\Omega \left(E_1(s,t), E_2(j,l)\right)^q + \beta |A_1(s,t) - A_2(j,l)|^q + (1-\alpha-\beta) d_\Psi \left(F_1(s), F_2(j)\right)^q \right]^p \pi_{s,l} \pi_{t,j} \right)^{\frac{1}{p}} \tag{42}$$

Since the permutation $\sigma$ is bijective from $[\![1, m_1]\!]$ to $[\![1, m_1]\!]$, we have $\mathcal{S} = [\![1, m_1]\!]$, which leads to

$$\min_{\pi \in \Pi(\boldsymbol{p}_1, \boldsymbol{p}_2)} \left( \sum_{s=1}^{m_1} \sum_{t=1}^{m_1} \sum_{j,l} \left[ \alpha d_\Omega \left(E_1(s,t), E_2(j,l)\right)^q + \beta |A_1(s,t) - A_2(j,l)|^q + (1-\alpha-\beta) d_\Psi \left(F_1(s), F_2(j)\right)^q \right]^p \pi_{s,l} \pi_{t,j} \right)^{\frac{1}{p}} \tag{43}$$

It's clear that Equation 43 describes the same minimization as the one of Equation 39, the proof is thus concluded. $\square$

## B  Additional Details on Algorithms

In this section, we present additional details on our algorithms.

---

**Algorithm 3** Line-Search in Conditional Gradient Descent

---

**Input:** $a$, $b$
**if** $a > 0$ **then**
    $\gamma^{(k)} = \min(1, \max(0, -\frac{2a}{b}))$
**else**
    **if** $a + b < 0$ **then**
        $\gamma^{(k)} = 1$
    **else**
        $\gamma^{(k)} = 0$
    **end if**
**end if**
**Output:** $\gamma^{(i)}$

---

### B.1 Line Search in Algorithm 1

Denoting $\Delta = \tilde{\pi}^{(k-1)} - \pi^{(k-1)}$, $\Lambda_1 = A^2 \boldsymbol{p} \mathbb{1}_{\tilde{m}}^{\mathsf{T}} + \mathbb{1}_m \tilde{\boldsymbol{p}}^{\mathsf{T}} \tilde{A}^{\mathsf{T}2}$ and $\Lambda_2 = g(E) \boldsymbol{p} \mathbb{1}_{\tilde{m}}^{\mathsf{T}} + \mathbb{1}_m \tilde{\boldsymbol{p}}^{\mathsf{T}} h(\tilde{E})^{\mathsf{T}}$, then we have $\pi^{(k)} = \pi^{(k-1)} + \gamma^{(k)} \Delta$ and

$$
\begin{aligned}
&\mathcal{E}_{\alpha,\beta}(\pi^{(k)}) \\
&= \left\langle (1 - \alpha - \beta) M(F, \tilde{F}) + \beta J(A, \tilde{A}) \otimes \pi^{(k)} + \alpha L(E, \tilde{E}) \otimes \pi^{(k)}, \pi^{(k)} \right\rangle \\
&= \left\langle (1 - \alpha - \beta) M(F, \tilde{F}) + \beta (\Lambda_1 - 2A\pi^{(k)} \tilde{A}^{\mathsf{T}}) + \alpha (\Lambda_2 - 2 \sum_{t=1}^{T} E[t] \pi^{(k)} \tilde{E}[t]^{\mathsf{T}}), \pi^{(k)} \right\rangle \\
&= \Big\langle (1 - \alpha - \beta) M(F, \tilde{F}) + \beta (\Lambda_1 - 2A(\pi^{(k-1)} + \gamma^{(k)} \Delta) \tilde{A}^{\mathsf{T}}) \\
&\quad + \alpha (\Lambda_2 - 2 \sum_{t=1}^{T} E[t] (\pi^{(k-1)} + \gamma^{(k)} \Delta) \tilde{E}[t]^{\mathsf{T}}), \pi^{(k-1)} + \gamma^{(k)} \Delta \Big\rangle
\end{aligned}
\tag{44}
$$

Then we can rewrite $\mathcal{E}_{\alpha,\beta}(\pi^{(k)})$ as $a\gamma^{(k)^2} + b\gamma^{(k)} + c$ with

$$
a = \left\langle -2\alpha \sum_{t=1}^{T} E[t] \Delta \tilde{E}[t]^{\mathsf{T}} - 2\beta A \Delta \tilde{A}^{\mathsf{T}}, \Delta \right\rangle
\tag{45}
$$

$$
\begin{aligned}
b &= \left\langle (1 - \alpha - \beta) M(F, \tilde{F}) + \alpha (\Lambda_2 - 2 \sum_{t=1}^{T} E[t] \pi^{(k-1)} \tilde{E}[t]^{\mathsf{T}}) + \beta (\Lambda_1 - 2A\pi^{(k-1)} \tilde{A}^{\mathsf{T}}), \Delta \right\rangle \\
&\quad + \left\langle -2\alpha \sum_{t=1}^{T} E[t] \Delta \tilde{E}[t]^{\mathsf{T}} - 2\beta A \Delta \tilde{A}^{\mathsf{T}}, \pi^{(k-1)} \right\rangle
\end{aligned}
\tag{46}
$$

Hence, the line-search presented in Algorithm 1 can be performed with Algorithm 3 using $a$ and $b$ as defined above.

### B.2 Graph Dictionary Learning with the FNGW Barycenter

We now describe a learning task that exploits the notion of FNGW-barycenter. Dictionary learning based on graphs is an extension of factorization methods to graphs and was popularized by the seminal works of Xu (2020) and Vincent-Cuaz et al. (2021). In our setting, given a set of labeled graphs $\{g_i\}_{i=1}^n$ with $g_i = (F_i, A_i, E_i, \boldsymbol{p}_i)$, we want to learn a dictionary of *atoms* $\{\overline{g}_s\}_{s=1}^S$ so that each graph of the training set can be reconstructed as a FNGW-barycenter of the dictionary atoms.

---
**Algorithm 4** Unmixing Problem Solver

---
    **input:** Graph $g$, Dictionary $\{\overline{g}_s\}_{s=1}^S$

    **init:** Initialize uniformly $\boldsymbol{w}$.

    **repeat**

        With $\boldsymbol{w}$ fixed, compute the barycenter with Algorithm 2, obtaining $\tilde{g} = \mathfrak{B}(\boldsymbol{w}, \{\overline{g}_s\}_{s=1}^S, \boldsymbol{p})$ and save $S$ independent optimal transport plans $\pi_s$ between $\tilde{g}$ and $\overline{g}_s$.

        Compute the optimal transport plan $\pi$ between $g$ and $\tilde{g}$.

        Compute the optimal $\boldsymbol{w}$ by minimizing Equation 48 with fixed transport plans $\pi$ and $\{\pi_s\}_s$ using the CG algorithm.

    **until** Convergence of the reconstruction loss.

    **output:** $\boldsymbol{w}$, $\pi$ and $\{\pi_s\}_s$

---

Assuming that the probability distributions for the atoms $\{\overline{\boldsymbol{p_s}}\}_{s=1}^S$ are fixed beforehand, our dictionary learning problem writes as:

$$\min_{\substack{\{\boldsymbol{w_i}\}_{i=1}^n s.t. \boldsymbol{w_i} \in \Sigma_S \\ \{(\overline{E}_s, \overline{A}_s, \overline{F}_s)\}_{s=1}^S}} \sum_{i=1}^n \mathrm{FNGW}_{\alpha,\beta}\Big(g_i, \mathfrak{B}(\boldsymbol{w_i}, \{\overline{g}_s\}_{s=1}^S, \boldsymbol{p_i})\Big) - \lambda \sum_i^n \|\boldsymbol{w_i}\|_2^2 \tag{47}$$

where the weights $\{\boldsymbol{w_i}\}_i$ describe the *embeddings* of our graphs in the dictionary and $\lambda$ is a regularization parameter which controls their sparsity.

**Remark B.1.** It should be noted that this problem setting corresponds to the dictionary learning in Xu (2020) rather than the one presented in Vincent-Cuaz et al. (2021), where a graph is projected on a linear representation of atoms. Here, representing a graph as a FNGW-barycenter of the atoms allows in particular to use atoms comprising various number of nodes.

We propose to solve the optimization problem of Equation 47 with a stochastic iterative procedure, which for each sampled batches of data alternatively updates the embeddings $\{\boldsymbol{w_i}\}_i$ and atoms $\{(\overline{E}_s, \overline{A}_s, \overline{F}_s)\}_{s=1}^S$. Finding the embedding $\{\boldsymbol{w_i}\}_i$ for a fixed dictionary requires an intermediate procedure called *unmixing*, which is in our case a bi-level optimization problem, since the reconstructed graph is the solution of a FNGW-barycenter problem. Assuming a small change in $\boldsymbol{w_i}$ will not affect the solution of the barycenter problem, we propose therefore to solve the latter first, and find the optimal $\boldsymbol{w_i}$ with the fixed OT plans of the barycenter.

Let us formalize the unmixing procedure, which can be described as solving the following problem:

$$\min_{\boldsymbol{w} \in \Sigma_S} \mathrm{FNGW}_{\alpha,\beta}\Big(g, \mathfrak{B}(\boldsymbol{w}, \{\overline{g}_s\}_{s=1}^S, \boldsymbol{p})\Big) - \lambda\|\boldsymbol{w}\|_2^2 \tag{48}$$

Our detailed algorithm is provided in Algorithm 4.

This procedure is applied to the graphs in $\{g_i\}_i$, giving us corresponding $\{\boldsymbol{w_i}\}_i$ and transport plans, which allows us to update atoms $\{(\overline{g}_s)\}_{s=1}^S$ by gradient descent. Given a batch of graph samples $\{g_b\}_{b=1}^B$, we define the following functional, representing the batch loss:

$$\mathfrak{L}(\overline{F}_s, \overline{A}_s, \overline{E}_s) = \frac{1}{B} \sum_{b=1}^B \mathcal{E}_{\alpha,\beta}\left((E_b, A_b, F_b), (\tilde{E}_b, \tilde{A}_b, \tilde{F}_b), \pi_b\right) \tag{49}$$

---

**Algorithm 5** Stochastic Graph Dictionary Learning

---

**input:** Graph dataset $\{g_i\}_{i=1}^n$
**init:** Randomly initialize the atoms $\{(\overline{E}_s, \overline{A}_s, \overline{F}_s)\}_{s=1}^S$
**for** $k = 1, \ldots, K$ **do**
    Sample a mini-batch of graphs from dataset: $\{(E_b, A_b, F_b, \boldsymbol{p}_b)\}_{b=1}^B$
    **for** $b = 1, \ldots, B$ **do**
        With $\{(\overline{E}_s, \overline{A}_s, \overline{F}_s)\}_{s=1}^S$ fixed, solve the unmixing problem: $\boldsymbol{w}_b, \pi_b, \{\overline{\pi}_{b,s}\}_s =$
Alg.4$((E_b, A_b, F_b, \boldsymbol{p}_b), \{(\overline{E}_s, \overline{A}_s, \overline{F}_s)\}_{s=1}^S)$
    **end for**
    **for** $s = 1, \ldots, S$ **do**
        With fixed $\boldsymbol{w}_b, \pi_b, \{\overline{\pi}_{b,s}\}_s$, compute the gradients of the mini-batch loss with respect to $\{\overline{E}_s, \overline{A}_s, \overline{F}_s\}$.
        Update $\{\overline{E}_s, \overline{A}_s, \overline{F}_s\}$ using gradients from Equation 53 to Equation 55.
    **end for**
**end for**
**Output:** The atoms of the dictionary $\{(\overline{E}_s, \overline{A}_s, \overline{F}_s)\}_{s=1}^S$

---

with $(\tilde{E}_b, \tilde{A}_b, \tilde{F}_b)$ the reconstructed graph of the batch sample $g_b$ from the results of the unmixing problem and $\pi_b$ the optimal transport plan between them. The reconstructed graph is then expressed as:

$$\tilde{E}_b = \frac{1}{\mathcal{I}_{m \times T} \times_2 \boldsymbol{pp}^\mathsf{T}} \sum_s \boldsymbol{w}_{s,b} (\overline{C}_s \times_2 \overline{\pi}_{s,b}) \times_1 \overline{\pi}_{s,b} \tag{50}$$

$$\tilde{A}_b = \frac{1}{\boldsymbol{pp}^\mathsf{T}} \circ \sum_s \boldsymbol{w}_{s,b} \overline{\pi}_{s,b} \overline{A}_s \overline{\pi}_{s,b}^\mathsf{T} \tag{51}$$

$$\tilde{F}_b = \sum_s \boldsymbol{w}_{s,b} \mathrm{diag}(\frac{1}{\boldsymbol{p}}) \overline{\pi}_{s,b} \overline{F}_s \tag{52}$$

where $\overline{\pi}_{s,b}$ denotes the optimal transport plan between the reconstructed graph $\tilde{g}_b$ and atom $\overline{g}_s$. Then the estimated gradients is written as:

$$\nabla_{\overline{F}_s} \mathfrak{L} = \frac{2}{B} \sum_{b=1}^B \boldsymbol{w}_{s,b} \overline{\pi}_{s,b}^\mathsf{T} \left( \tilde{F}_b - \mathrm{Diag}(\frac{1}{\boldsymbol{p}_b}) \pi_b^\mathsf{T} F_b \right) \tag{53}$$

$$\nabla_{\overline{A}_s} \mathfrak{L} = \frac{2}{B} \sum_{b=1}^B \boldsymbol{w}_{s,b} \overline{\pi}_{s,b}^\mathsf{T} \left( \tilde{A}_b - \frac{1}{\boldsymbol{p}_b \boldsymbol{p}_b^\mathsf{T}} \circ \pi_b^\mathsf{T} A_b \pi_b \right) \overline{\pi}_{s,b} \tag{54}$$

$$\nabla_{\overline{E}_s} \mathfrak{L} = \frac{2}{B} \sum_{b=1}^B \boldsymbol{w}_{s,b} \left( \left( \tilde{E}_b - \frac{1}{\mathcal{I}_{m \times T} \times_2 \boldsymbol{p}_b \boldsymbol{p}_b^\mathsf{T}} \circ \left( E_b \times_2 \pi_b^\mathsf{T} \right) \times_1 \pi_b^\mathsf{T} \right) \times_2 \overline{\pi}_{s,b}^\mathsf{T} \right) \times_1 \overline{\pi}_{s,b} \tag{55}$$

The complete algorithm for dictionary learning is summarized by Algorithm 5.

## C  Additional Details on Experiments

This section is dedicated to additional details about the experiments carried out in Section 4.

### C.1  Edge-featured Graph Classification

**Datasets.** We consider 7 graph datasets for classification: Cuneiform (Kriege et al., 2018), MUTAG (Debnath et al., 1991; Kriege & Mutzel, 2012), PTC-MR (Helma et al., 2001; Kriege & Mutzel, 2012), BZR-MD, COX2-MD, DHFR-MD and ER-MD (Sutherland et al., 2003; Kriege & Mutzel, 2012). While Cuneiform is one of the earliest systems of writing realized by wedge-shaped marks on clay tablets, the rest of the datasets are of the nature of small molecules. Table 4 describes the details of features occurring in these datasets. All the datasets can be downloaded from `https://ls11-www.cs.tu-dortmund.de/staff/morris/graphkerneldatasets`.

Table 4: The details of the features concerning the datasets used in graph classification.

| Features | Cuneiform | MUTAG | PTC-MR | BZR-MD | COX2-MD | DHFR-MD | ER-MD |
|---|---|---|---|---|---|---|---|
| Node label (Discrete) | Yes | Yes | Yes | Yes | Yes | Yes | Yes |
| Node attribute (Real-valued) | Yes | No | No | No | No | No | No |
| Edge label (Discrete) | Yes | Yes | Yes | Yes | Yes | Yes | Yes |
| Edge attribute (Real-valued) | Yes | No | No | Yes | Yes | Yes | Yes |

Table 5: The types of the features treated by the methods used in graph classification.

| Features | RWK | SPK | GK | WLK-VH | HOPPERK | PROPAK | FGW | NSPDK | EHK | VEHK | WLK-VEH | PNA | GAT | FNGW |
|---|---|---|---|---|---|---|---|---|---|---|---|---|---|---|
| Node label (Discrete) | Yes | Yes | No | Yes | Yes | Yes | Yes | Yes | No | Yes | Yes | Yes | Yes | Yes |
| Node attribute (Real-valued) | No | No | No | No | No | No | Yes | No | No | No | No | Yes | Yes | Yes |
| Edge label (Discrete) | No | No | No | No | No | No | No | Yes | Yes | Yes | Yes | Yes | Yes | Yes |
| Edge attribute (Real-valued) | No | No | No | No | No | No | No | No | No | No | No | Yes | Yes | Yes |

**Experimental Settings.** For datasets whose node features are discrete, as in Vayer et al. (2020), we use Weisfeiler-Lehman (WL) labeling to transform them into vectors of dimension $K$, being the number of iterations in WL labeling. We use a dataset-specific random vector as feature for empty edges. To compute the FNGW distance, regarding the distance $d_\Psi$, we use the Hamming distance when WL labeling is used, or the $\ell^2$-norm otherwise. The distance $d_\Omega$ between edge features is the $\ell^2$-norm in all cases.

For all methods, we conduct nested cross-validation with 50 iterations of the outer CV loop and 10-fold inner CV. In each outer CV loop, the dataset is split into a training set and a test set, with 10% of the data held out for the latter. The splitting is kept consistent across all methods for a fair comparison. We report the mean and standard deviation of the test accuracy.

For the classifiers based on FNGW or FGW, the coefficient $\gamma$ presented in the Gram matrix computation is cross-validated within $\{2^i \mid i \in [\![-10, 10]\!]\}$ while the number of iteration $K$ in WL labeling is searched in $[\![0, 4]\!]$ for MUTAG and PTC-MR or $[\![0, 3]\!]$ for the other datasets (0 means no WL labeling is used) except Cuneiform. For FNGW, we cross-validate 7 values of the trade-off parameter $\alpha$ via a logspace search in $[0, 0.5]$ and the same strategy is applied to $\beta$, while a total of 15 values are drawn from logspaces $[0, 0.5]$ and $[0.5, 1]$ to cross-validate $\alpha$ for FGW. For the kernel-based methods, decay factor $\lambda$ of RWK is cross-validated from $\{10^i \mid i \in [\![-6, -2]\!]\}$, the number of iterations of WLK is cross-validated in $[\![1, 10]\!]$ while the one of PROPAK is chosen from $\{1, 3, 5, 8, 10, 15, 20\}$. For GK, the CV range of the graphlet size is $\{3, 4, 5\}$ and the one of the precision level $\epsilon$ and the confidence $\delta$ is $\{0.1, 0.05\}$. For NSPDK, the maximum considered radius $r$ between vertices is cross-validated within $[\![0, 5]\!]$, and neighborhood depth $d$ is chosen from $\{3, 4, 5, 6, 8, 10, 12, 14, 20\}$. Finally, the regularization coefficient $C$ of the SVM is cross-validated within $\{10^i \mid i \in [\![-7, 7]\!]\}$ for all the methods except for the case MUTAG-FNGW where the value $10^7$ is not included.

**Runtime Comparison.** We compare the Gram matrix computation time of the different graph kernels, FGW and FNGW used in the experiments. The results are presented in Table 6.

## C.2  Fingerprint to Molecule.

**Experimental Settings.** Table 7 outlines the details of the hyper-parameter searching range for the cross-validation for both FGW and FNGW loss. The constrained exploration of the NNBary range reflects the substantial computational resources demanded by this method.

## C.3  Metabolite Identification.

**Dataset.** To evaluate the performance for metabolite identification from tandem mass spectra, we use the data extracted and processed in Dührkop et al. (2015), which is a set of 4,022 small compounds from the public GNPS Public Spectral Libraries (`https://gnps.ucsd.edu/ProteoSAFe/libraries.jsp`). The candidate sets were built with molecular structures from PubChem. The dataset can be downloaded from `https:`

Table 6: Runtime of Gram matrix computation for graph classification on MUTAG, PTC-MR, and BZR-MD datasets. 10 times of computation are conducted for each method. The last five rows show the results for methods leveraging the edge features.

| Methods | MUTAG | PTC-MR | BZR-MD |
|---------|-------|--------|--------|
| RWK | $38.756 \pm 0.126$ | $64.095 \pm 0.473$ | $114.663 \pm 0.646$ |
| SPK | $0.283 \pm 0.019$ | $0.431 \pm 0.019$ | $1.423 \pm 0.078$ |
| GK | $42.201 \pm 0.317$ | $78.518 \pm 0.124$ | $83.448 \pm 0.122$ |
| WLK-VH | $0.072 \pm 0.006$ | $0.089 \pm 0.011$ | $0.385 \pm 0.016$ |
| HOPPERK | $11.491 \pm 0.048$ | $46.148 \pm 0.549$ | $14.411 \pm 0.070$ |
| PROPAK | $0.344 \pm 0.033$ | $0.796 \pm 0.026$ | $0.633 \pm 0.016$ |
| FGW | $66.120 \pm 0.202$ | $167.880 \pm 0.209$ | $99.201 \pm 0.152$ |
| NSPDK | $1.736 \pm 0.019$ | $2.242 \pm 0.040$ | $34.017 \pm 0.259$ |
| EHK | $0.002 \pm 0.000$ | $0.003 \pm 0.000$ | $0.022 \pm 0.004$ |
| VEHK | $0.005 \pm 0.000$ | $0.008 \pm 0.001$ | $0.069 \pm 0.019$ |
| WLK-VEH | $0.076 \pm 0.002$ | $0.120 \pm 0.009$ | $0.731 \pm 0.010$ |
| FNGW | $240.508 \pm 4.958$ | $565.021 \pm 8.726$ | $881.524 \pm 27.510$ |

Table 7: Hyper-parameter searching ranges during the cross-validation on Fing2Mol dataset.

| Hyper-parameters | NNBary | ILE |
|------------------|--------|-----|
| $\alpha$ of the FGW/FNGW | $\{0.01, 0.5, 0.9\}$ | $\{0.01, 0.1, 0.5, 0.9, 0.99\}$ |
| Number of graphs for barycenter | $\{10, 15\}$ | $\{15, 20, 25\}$ |
| $\gamma$ of the Gaussian input kernel | - | $\{10^{-6}, 10^{-5}, 10^{-4}, 10^{-3}, 10^{-2}, 10^{-1}\}$ |
| Ridge regularization parameter $\lambda$ | - | $\{10^{-10}, 10^{-8}, 10^{-6}, 10^{-4}, 10^{-2}\}$ |
| Sketching size | - | $\{1000, 5000\}$ |
| Hidden dimension of MLP | $\{128\}$ | - |

//zenodo.org/record/804241#.Yi9bzS_pNhE, which is released under *Creative Commons Attribution 4.0 International* license.

**Experimental Settings.**   We choose the ridge regularization parameter ($\lambda = 10^{-4}$) and the diffusion rate ($\tau = 0.6$) following Brogat-Motte et al. (2022). We use a separate validation set (1/5 of the training set) to select the hyperparameters $\alpha$ and $\beta$ of FNGW loss. $\alpha$ and $\beta$ are chosen from $\{0.1, 0.33, 0.5, 0.67\}$ with the constraint $\alpha + \beta < 1$. For the experiment with FGW, we keep the same hyper-parameter as in Brogat-Motte et al. (2022).

# D   Additional Experiments

In this section, we present additional experiments showing the application of our proposed FNGW distance.

## D.1   Runtime of FNGW Computation with Respect to Graph Size

To give an idea of the scales involved, the run-time with respect to the number of nodes (the graphs are randomly created with both node feature and edge feature of dimension 2; the graphs to be compared are of the same size) is shown in Table 8. The maximum number of CGD steps $K$ is set to 100 for both distances. The experiments are conducted on an Intel(R) Xeon(R) CPU E5-2650 v2. While experiments in this paper are made on small graphs, where the difference in computation time between FGW and FNGW is reasonable, applying FNGW efficiently on large graphs will require further investigation.

Table 8: Runtime ($s$) of computation of the FNGW w.r.t graph size.

| # nodes | FNGW | FGW |
|---|---|---|
| 25 | $0.0294 \pm 0.0089$ | $0.0039 \pm 0.0003$ |
| 50 | $0.0949 \pm 0.0290$ | $0.0047 \pm 0.0009$ |
| 100 | $1.4640 \pm 0.5598$ | $0.0137 \pm 0.0005$ |
| 200 | $10.2419 \pm 2.8562$ | $0.0435 \pm 0.0063$ |

Table 9: Computation runtime ($s$) comparison of FNGW and GED on QM9 molecules.

| FNGW ($\alpha = 0.33$, $\beta = 0.33$) | GED w/ Edge feature |
|---|---|
| $0.63 \pm 0.01$ | $276.84 \pm 0.56$ |

## D.2   Runtime Comparison Between FNGW and GED

We randomly select 100 pairs of molecules from the QM9 dataset and compute the distance between each pair using FNGW or GED. This procedure is repeated 10 times, and the total time required is reported in Table 9. We utilize the NetworkX implementation of GED.

## D.3   Real Migration Network Clustering

In this experiment, we use the global bilateral migration networks released by the World Bank and processed by Chowdhury & Mémoli (2019) to test our distance. Differently than Chowdhury & Mémoli (2019), the capacity of the FNGW distance to model higher dimensional edge features allows us to take in account both networks (corresponding to the male migration and the female migration, separated) conjointly, by combining their edge features. The resulting dataset consists of 5 graphs, each containing 225 nodes representing countries or administrative regions. The feature of each edge $(i \rightarrow j)$ is a two-dimensional vector, where each indicates the number of respectively male and female migrating from region $i$ to region $j$. Since the networks are complete and do not possess any node features, the trade-off parameter $\alpha$ of the FNGW distance is set to 1. The weights of the nodes in the network are uniformly distributed. Figure 7 shows the dissimilarity matrix of the migration networks and its associated single-linkage clustering dendrogram.

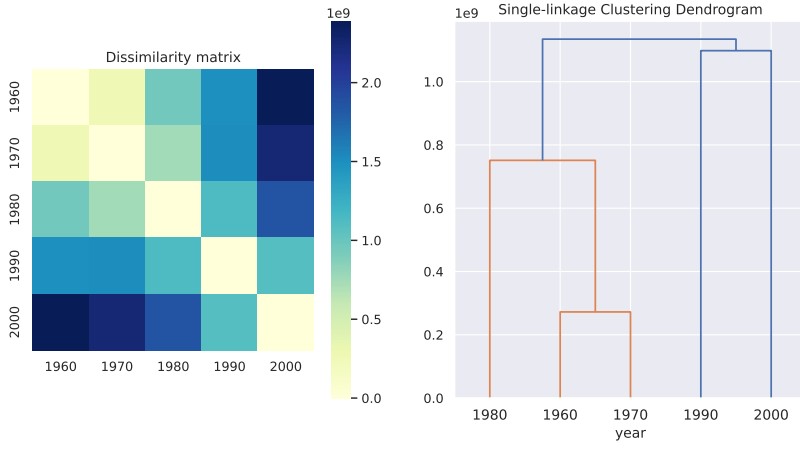

Figure 7: Results of migration networks

### D.4 Synthetic Labeled Graph Clustering

In this synthetic task, we generate 3 groups of graphs, following the Stochastic Block Model (SBM) with the number of blocks in each graph chosen from $\{1, 2, 4\}$. For each cluster, we generate 15 graphs whose numbers of nodes are randomly chosen from $\{20, 30, 40\}$. For each graph, the feature of the nodes from block $i$ is sampled from $\mathcal{N}(i, 1)$. For the edges, we consider the following 3 labels: {black, blue, green}. Edges between nodes in the same block are labeled as black with probability 1. For every pair of nodes $(p, q)$ with $p$ from block $i$ and node $q$ from block $i + 1$, there exists an edge $p \rightarrow q$ labeled as green, and there exists an edge $q \rightarrow p$ labeled as blue with the same probability. There is no other possible edges. In total, the dataset contains 45 graphs. Several synthetic graph samples are shown in Figure 8. We apply the $k$-means algorithm in order to perform graph clustering, considering the FNGW barycenter as the centroid of each cluster and the FNGW distance as the cluster assignment metric. Cluster centroids are randomly initialized with 20 nodes; their evolution as found by $k$-means is presented in Figure 9. We can observe that the resulting centroids recover not only the node features but also the characteristic of the edge labels of the different groups in the synthetic dataset.

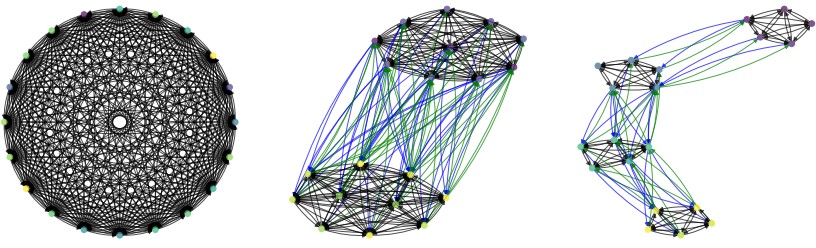

Figure 8: Samples of graphs for clustering with 20 nodes

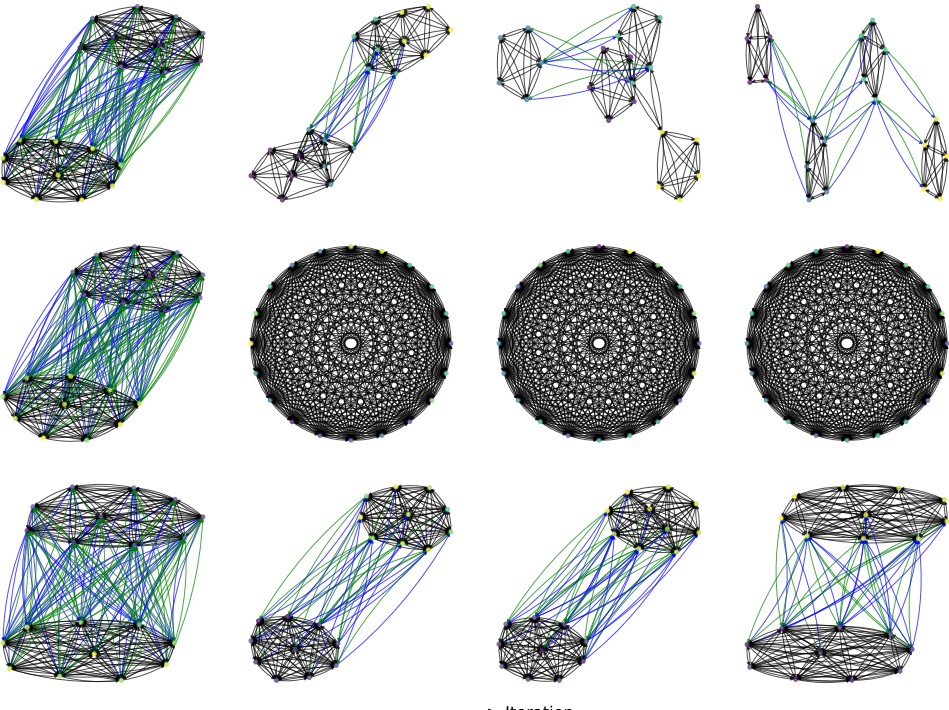

-----------------> Iteration

Figure 9: Evolution of centroids of each cluster found by $k$-means algorithm.

### D.5 Representation Learning through Dictionary Learning

We structure this toy problem around 3 graph templates, that we generate following the Stochastic Block Model (SBM) with 1, 2 and 3 blocks. Each template consists of 20 nodes. For each, node features are scalars representing the block they belong to. Within a block, all nodes are connected with black edges. Pairs of nodes from adjacent blocks are connected with probability 0.3 by edges that are colored in blue when ascending, green when descending. The Templates are shown in Figure 10a. We repeat the following procedure to create a synthetic dataset for dictionary learning: given a vector $v \in \mathbb{R}^3$ which components are sampled from $\mathcal{N}(0, 1)$, we compute a simplex $w = \text{softmax}(v)$ and generate a new graph by computing the barycenter of the templates defined above with Algorithm 2. The number of nodes is set to be 20 for all barycenters. To control the quality of the generated barycenters, we set a maximum barycenter loss value of 0.3. The goal of the experiment is to retrieve graphs presenting the same characteristics with the 3 original templates through dictionary learning, even with a varying number of nodes. Setting the numbers of nodes of the atoms to 5, 10 and 15 respectively, we perform dictionary learning on the dataset. The learned atoms are shown in Figure 10b, and clearly recover the properties of the synthetic templates.

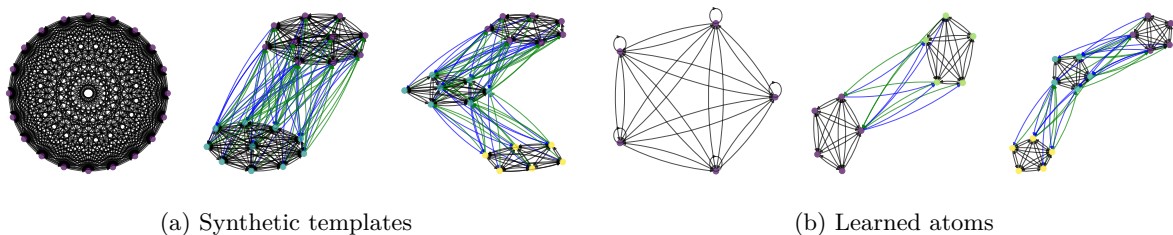

(a) Synthetic templates                        (b) Learned atoms

Figure 10: Dictionary learning on synthetic dataset.

### D.6 More Prediction Examples on Fin2Mol Dataset.

Figure 11 shows more predictions of the different methods on Fin2Mol dataset.

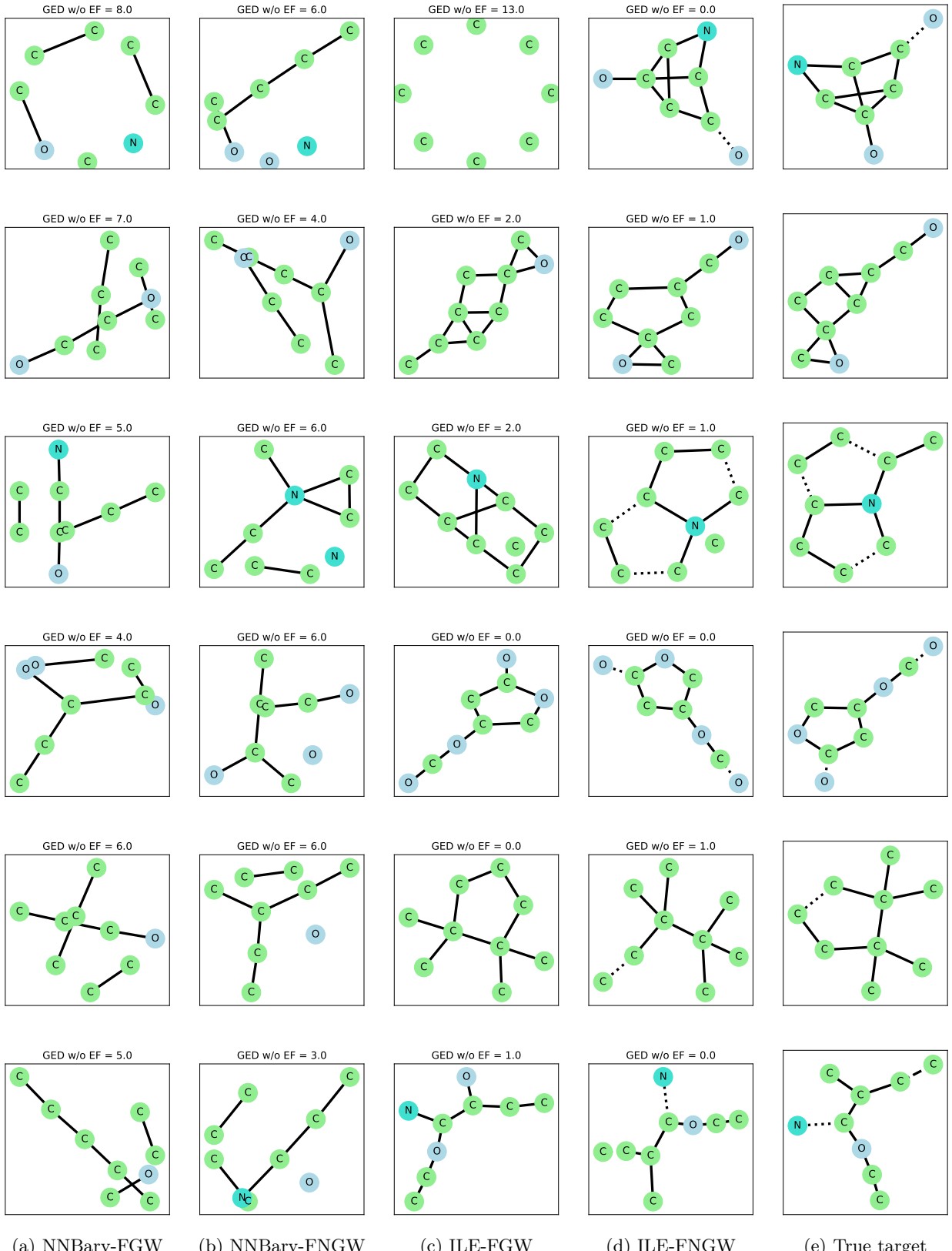

Figure 11: More predicted molecules on the Fin2Mol dataset.

