# OpenReview forum: "Exploiting Edge Features in Graph-based Learning with Fused Network Gromov-Wasserstein Distance"
_TMLR — Accepted by TMLR_

### Review · Reviewer_EgyT · 2024-05-05

**Summary Of Contributions:**

The paper proposes an extension of fused network Gromov-Wasserstein distance so that edge attributes can be incorporated into the distance. The cost matrix of the transportation is defined by a linear combination of edge attribute distance, difference in the shortest path matrix, and the distance of node attributes. Further, an application to the barycenter problem and its supervised prediction are also discussed. The barycenter optimization is formulated as a block coordinate descent procedure, and the framework of the supervised graph prediction is also discussed.

**Audience:**

Yes

**Claims And Evidence:**

Yes

**Requested Changes:**

- Recently, the fused Gromov Wasserstein with edge-attribute has been consider by the following paper. Although it is quite recent, relation should be clarified in detail for scientific fairness.
  - K. Kawano, S. Koide, H. Shiokawa, T. Amagasa, Multi-dimensional Fused Gromov Wasserstein Discrepancy for Edge-Attributed Graphs, IEICE TRANSACTIONS on Information and Systems, 2024


- Theorem 2.6 should be explained in more detail. In particular, what the last two properties indicate? The last one is similar to the triangular inequality, but slightly different. In the end, does FNGW satisfy the axiom of metric?


- The third property of theorem 2.6 guarantees FNGW(g_X,g_X) = 0. However, because of the non-convexity of the FNGW optimization problem, it is not clear whether this condition can be satisfied in practice. This seems also same for The second and last properties. I hope it clarified.

- Although in the experiments, exp(-gamma FNGW(g_i,g_i)) is used, is it guaranteed to be positive semi-definite? (I guess it isn't). It should be clarified that how it is handled.

- GED in (16) does not depend on edge attributed? If so, how does the edge attribute contribute to the prediction?

- In Appendix D.1 the authors mentioned 'the difference in computation time between FGW and FNGW is negligible'. However, I do not agree with this claim because the computational time of FNGW and FGW has 10 times difference even in #nodes=25. Since kernel method is used in the experiments, at least, O(train_size^2) times distance calculations are requires by which I guess difference is not negligible.

- The authors mentioned that hyper-parameters are selected by CV. Does that mean the nested CV is performed? (hyper-parameters are selected by the inner CV, and table 1 is performance on the outer CV?).

**Strengths And Weaknesses:**

S: The problem setting should be important for the community, and the experimental results indicate high prediction performance.

W: Some technical detail should have been clarified in more detail as I mentioned below.

W: I don't fully understand the novelty of the paper. The edge attribute simply introduced through the distance of the attribute, and the optimization applies the Frank-Wolfe algorithm as is. Supervised graph prediction is directly taken from Ciliberto et al., 2020, and Brogat-Motte et al., 2022. In other words, technical difficulty newly solved by this paper is not clear (Note that I'm aware of that technical significance is not necessarily required in TMLR).

---

> ### Author Response · Authors · 2024-05-24
> **Response to Reviewer EgyT**
>
> We would like to thank the reviewer for their comments and suggestions. We did our best to answer and revise the paper consequently.
>
> > I don't fully understand the novelty of the paper. The edge attribute simply introduced through the distance of the attribute, and the optimization applies the Frank-Wolfe algorithm as is. Supervised graph prediction is directly taken from Ciliberto et al., 2020, and Brogat-Motte et al., 2022. In other words, technical difficulty newly solved by this paper is not clear (Note that I'm aware of that technical significance is not necessarily required in TMLR).
>
> **Response:**  While FNGW can be seen as a natural extension of FGW, obtained by introducing an inner distance between edge features, we provide a comprehensive theoretical analysis to establish the metric properties of FNGW in the general case (see Theorem 2.8 and Appendix A.2). Moreover, Proposition 2.9 provides a novel property regarding edge features of barycenters. Concerning supervised graph prediction, Ciliberto et al. (2020) provide a nice theoretical framework, ILE, for structured prediction, but the practical applications presented in the paper are quite limited. Brogat-Motte et al. (2022) apply ILE to supervised graph prediction with FGW as the loss function. Our work differs from theirs in the following aspects:
> 1. FNGW, which we prove satisfies the ILE condition, enables us to predict the edge features of graphs, which is not possible in the work of Brogat-Motte et al. (2022).
> 2. Brogat-Motte et al. (2022) constrain their experiments with ILE-FGW to situations where a candidate set is provided, while in our introduced fingerprint-to-molecule experiment (see Section 4.2), the prediction is done in an end-to-end fashion, allowing for the prediction of entirely new structures.
> 3. We extend ILE to the big data regime with sketching, which enables us to solve structure prediction problems on more than 100,000 training data points with kernel methods.
>
> > Recently, the fused Gromov Wasserstein with edge-attribute has been consider by the following paper. Although it is quite recent, relation should be clarified in detail for scientific fairness. (K. Kawano, S. Koide, H. Shiokawa, T. Amagasa, Multi-dimensional Fused Gromov Wasserstein Discrepancy for Edge-Attributed Graphs, IEICE TRANSACTIONS on Information and Systems, 2024)
>
> **Response:** We thank the reviewer for pointing out this reference that we were not aware of. A first version of the work we present here was first submitted at NeurIPS 2023 on May 23, 2023, and later publicly available on arXiv on sept 28, 2023. We note that the reference mentioned by the reviewer does not cite our arxiv submission and we are puzzled by some similarities with our original submission, notably the choice of datasets and figures. Nevertheless, the nature of Kawano et al.'s paper seems to us rather different from a Machine Learning point of view: in our paper, each novel notion comes with an algorithm to compute it. We also rely on a broader view of FNGW properties offering deeper insights. Moreover, our goal is not to just introduce this extension of FGW distance but also to emphasize its utility in an involved learning task such as Structured output prediction (graph prediction). In contrast to our paper,the short paper in EICE focuses on classification tasks and subgraph matching, also does not study the metric properties in the general case. We added a note to refer to this recent article (section 2.1).
>
> > Theorem 2.6 should be explained in more detail. In particular, what the last two properties indicate? The last one is similar to the triangular inequality, but slightly different. In the end, does FNGW satisfy the axiom of metric?
>
> **Response:** Thanks for the suggestion. The third property, in fact, defines a weak isomorphism relation between objects of $\mathcal{G}$, and it guarantees that the FNGW distance between $g_X$ and $g_Y$ is 0 if and only if $g_X$ is weakly isomorphic to $g_Y$. The last one is actually a relaxed triangle inequality with a factor of $2^{q-1}$. When $q=1$, the triangle inequality is well satisfied.  We have provided a more detailed explanation of Theorem 2.6 (Theorem 2.8 in new version) (see the blue text around Theorem 2.8). In summary, FNGW satisfies the positivity, the equality, the symmetry, and the triangle inequality when $q=1$.

---

> > ### Author Response · Authors · 2024-05-24
> > **Response to Reviewer EgyT**
> >
> > **Response:** You are right. As other GW-based distances (they involve an non-convex optimization problem), due to the limitation of the optimization algorithm, the metric properties of FNGW can not always be guaranteed. We have made this point clear in the new version of the paper (see the blue text after Theorem 2.8).
> >
> > > Although in the experiments, exp(-gamma FNGW(g_i,g_i)) is used, is it guaranteed to be positive semi-definite? (I guess it isn't). It should be clarified that how it is handled.
> >
> > **Response:** As a matter of fact, $\exp(-\gamma FNGW(g_i,g_j))$ is not guaranteed to be positive semidefinite since itself involves a minimization problem. Different ways have been developed to handle this case, through dedicated regularization (see Luss et D'aspremont, 2007) or through stabilization (Ong et al., 2004). However, in this paper, on the datasets we considered, we did not need to apply these schemes and apply directly standard SVM algorithm.  We have clarified this in the revised paper.
> > * Ong, C. S., Mary, X., Canu, S., & Smola, A. J. (2004). Learning with non-positive kernels. Proceedings of the Twenty-First International Conference on Machine Learning, 81.
> > * Luss, R., & D’ aspremont, A. (2007). Support Vector Machine Classification with Indefinite Kernels. In J. Platt, D. Koller, Y. Singer, & S. Roweis (Eds.), Advances in Neural Information Processing Systems (Vol. 20).
> >
> >
> >
> > > GED in (16) does not depend on edge attributed? If so, how does the edge attribute contribute to the prediction?
> >
> > **Response:** GED in Equation (16) (Equation 18 in new version) depends on edge features since we consider edge substitution (i.e., changing the label of a given edge) as one of the edit operations. Please refer to the text below Equation (18).
> >
> >
> >
> > > In Appendix D.1 the authors mentioned 'the difference in computation time between FGW and FNGW is negligible'. However, I do not agree with this claim because the computational time of FNGW and FGW has 10 times difference even in #nodes=25. Since kernel method is used in the experiments, at least, O(train_size^2) times distance calculations are requires by which I guess difference is not negligible.
> >
> > **Response:** You are right, thanks for the remark. We have corrected our claim on this point (see blue in Appendix D.1). You can also find the comparison of FNGW and FGW in terms of Gram matrix computation time in Table 6. **We leave the task of developing the acceleration algorithm for future work.**
> >
> > > The authors mentioned that hyper-parameters are selected by CV. Does that mean the nested CV is performed? (hyper-parameters are selected by the inner CV, and table 1 is performance on the outer CV?).
> >
> >
> > **Response:** Yes, we performed nested CV on the graph classification task. The result in Table 1 is estimated by averaging accuracies over 50 outer test sets. We have clarified this point in the revised paper (See p.8 in blue). Thank you for pointing it out.

---

### Review · Reviewer_WuDS · 2024-05-12

**Summary Of Contributions:**

The authors propose a new distance for graphs based on optimal transport. They extend the Fused Gromov-Wasserstein Distance to the case of graphs with edge labels. They additionally provide an algorithm to compute the barycentre of a set of graphs based on the proposed distance notion.

**Audience:**

Yes

**Claims And Evidence:**

Yes

**Requested Changes:**

1. It would be of interest to see how the proposed approach performs on the other application domains: NLP, CV and temporal graphs.
2. Preface each theorem and symbol with an explanation of its meaning in English.
3. An explanation of how the distance computation is not affected by isomorphic permutations would be beneficial (or showing what is the effect of such permutations on the distance).
4. Please add a clarification on the treatment of real valued edge attributes. Please add take away message in captions.
5. It should be possible to conceive a clear artificial case where the result of the barycentre computation is known. For example one could iteratively perturb a graph (e.g. using operators such as node addition/removal or edge addition/removal) to generate a set of graphs such that the original graph would be the average/barycentre and compare the computed barycentre graph to the source graph.
6. It should be of interest to compare the correlation score of the proposed approach to the other similarity measures used in 4.1.1.

Critical adjustments are: 2 and 5

**Strengths And Weaknesses:**

1. The premise is that graphs with edge attributes constitute an important application case. The examples given are: 1) abstract meaning representation in NLP, 2) scene graphs in computer vision, 3) cheminformatics and 4) temporal graphs where the edge features evolve over time. However the authors offer an empirical evaluation of their approach only in the cheminformatics domain.
2. The introduction of the approach is overly formal, hard to follow and lacks a natural language guiding: section 2 needs to preface each theorem and each formal entity with a description in plain English (what are the concepts used in Eq. 9? what are \phi_X \psi_X \omega_X ?)  , what is theorem 2.5 and 2.6 saying in English? What is the interesting property for the barycentre enunciated in Proposition 2.9 in English?
3. The formulation of the approach makes use of the matrices E and A that code the graph structure in a non invariant way, i.e. not invariant to an isomorphic permutation of the node ordering.
4. Clarifications. The empirical evaluation needs to clarify if real valued edge attributes are taken into consideration, in this case please be clear on: 1) datasets properties, i.e. do graphs have only discrete edge labels or also real valued edge attributes? ad 2) methods capabilities, i.e. can the method  process discrete edge labels?  can it process real valued edge attributes?
Please include a takeaway message in the caption of the figures, for example in Figure 2, what does the reader need to understand from the two rightmost plots? that there is an optimal value for alpha and beta and that it is robust/not-robust?
5. The barycentre computation is the second major contribution of the paper and deserves: 1) a more detailed justification of why it is an important operation and how it can be used in applications and 2) a more intuitive and convincing empirical demonstration.
6. The edit distance correlation measure is a simple to understand quality score. However the score is not compared to other approaches.

---

> ### Author Response · Authors · 2024-05-24
> **Response to Reviewer WuDS**
>
> We would like to thank the reviewer for their comments and suggestions. We did our best to answer and revise the paper consequently.
>
> > The premise is that graphs with edge attributes constitute an important application case. The examples given are: 1) abstract meaning representation in NLP, 2) scene graphs in computer vision, 3) cheminformatics and 4) temporal graphs where the edge features evolve over time. However the authors offer an empirical evaluation of their approach only in the cheminformatics domain.
>
> > It would be of interest to see how the proposed approach performs on the other application domains: NLP, CV and temporal graphs.
>
> **Response:** Currently, while we are primarily focused on applications within the realm of cheminformatics, we also experiment on Cuneiform. This dataset comes from the computer vision domain, which also offers a promising field of application for our proposed FNGW. We believe that other examples are a good motivation for our work, and would also be interesting for applying FNGW, but we choose to leave them for future works, as the representation and transformation of features for those domains may represent sizable work, outside of the scope of our current paper.
>
> > The introduction of the approach is overly formal, hard to follow and lacks a natural language guiding: section 2 needs to preface each theorem and each formal entity with a description in plain English (what are the concepts used in Eq. 9? what are \phi_X \psi_X \omega_X ?) , what is theorem 2.5 and 2.6 saying in English? What is the interesting property for the barycentre enunciated in Proposition 2.9 in English?
>
> > Preface each theorem and symbol with an explanation of its meaning in English.
>
> **Response:** Thank you for the suggestion. We have clarified our theoretical statements in the revised paper. Please refer to the blue text after Eq.9, the changes (in blue) in Definition 2.1 and 2.3 (accompanied by Examples 2.2 and 2.4), addition before and within Theorems 2.7 and 2.8, and the remark after Proposition 2.11.
>
> > The formulation of the approach makes use of the matrices E and A that code the graph structure in a non invariant way, i.e. not invariant to an isomorphic permutation of the node ordering.
>
> > An explanation of how the distance computation is not affected by isomorphic permutations would be beneficial (or showing what is the effect of such permutations on the distance).
>
> **Response:** Indeed, the matrices $E$ and $A$ are encoded in a non-invariant way, but FNGW is not affected by isomorphic permutations of graphs. We have added a proof of this result in the Appendix A.7 of the revised version.
>
> > Clarifications. The empirical evaluation needs to clarify if real valued edge attributes are taken into consideration, in this case please be clear on: 1) datasets properties, i.e. do graphs have only discrete edge labels or also real valued edge attributes? ad 2) methods capabilities, i.e. can the method process discrete edge labels? can it process real valued edge attributes?
> Please include a takeaway message in the caption of the figures, for example in Figure 2, what does the reader need to understand from the two rightmost plots? that there is an optimal value for alpha and beta and that it is robust/not-robust?
>
> > Please add a clarification on the treatment of real valued edge attributes. Please add take away message in captions.
>
> **Response:** For the classification task, the details of dataset properties can be found in Table 4. In the revised version, we have also added details of the types of features treated by the methods in Table 5. For the Fingerprint to Molecule task, no real-valued edge attributes are present; the edge features concern only the type of bonds, as stated in the paper. For Metabolite Identification, ILE-FNGW utilizes the bond type and bond stereochemistry, both of which are discrete (one-hot encoding). Necessary take-away messages have been added to the captions. Thank you for the suggestion.
>
>
> > The barycentre computation is the second major contribution of the paper and deserves: 1) a more detailed justification of why it is an important operation and how it can be used in applications and 2) a more intuitive and convincing empirical demonstration.
>
> > It should be possible to conceive a clear artificial case where the result of the barycentre computation is known. For example one could iteratively perturb a graph (e.g. using operators such as node addition/removal or edge addition/removal) to generate a set of graphs such that the original graph would be the average/barycentre and compare the computed barycentre graph to the source graph.
>
> **Response:** We fully agree about the importance of barycenter computation. A more comprehensive justification has been provided at the end of Section 2. In addition to the barycenter computation example already given in Section 4.1.4, we have added another example in Figure 3, as suggested.

---

> > ### Author Response · Authors · 2024-05-24
> > **Response to Reviewer WuDS**
> >
> > > The edit distance correlation measure is a simple to understand quality score. However the score is not compared to other approaches.
> >
> > > It should be of interest to compare the correlation score of the proposed approach to the other similarity measures used in 4.1.1.
> >
> > **Response:** Graph Edit Distance (GED) is an important method for measuring the similarity between pairs of graphs in an error-tolerant manner (Gao et al., 2010). We chose GED as a benchmark metric to compare the performance of different supervised graph prediction methods. Measuring the correlation score between FNGW and GED helps in selecting the parameters of FNGW, as we aim for FNGW, which is differentiable, to approximate GED as a loss function. If we had initially considered the graph kernel as the performance metric, we could have conducted the same correlation procedure, but that is not the case in our experiment.
> >
> > * Gao, X., Xiao, B., Tao, D. et al. A survey of graph edit distance. Pattern Anal Applic 13, 113–129 (2010).

---

### Review · Reviewer_ZFM9 · 2024-05-13

**Summary Of Contributions:**

In measuring the distance between graphs, this paper proposes incorporating not only node features but also edge features of graphs using the Gromov-Wasserstein distance, which is applicable to downstream graph machine learning tasks. The proposal extends the fused and network Gromov-Wasserstein distance, called FNGW, and its theoretical properties as a metric have been analyzed. Furthermore, its effectiveness in graph classification and prediction tasks has been evaluated on real-world graph datasets, compared to other established methods such as graph kernels and graph neural networks.

**Audience:**

Yes

**Broader Impact Concerns:**

I do not have any concerns.

**Claims And Evidence:**

No

**Requested Changes:**

Please address the concerns outlined in the Weaknesses section above.

**Strengths And Weaknesses:**

### Strength
- This paper addresses the challenge of effectively incorporating edge features within the framework of OT-based distance for graphs. This method is considered to be widely applicable, and also can be a good literature for further development of this line of research.
- The proposal has been generalized to bounded continuous measurable functions on Polish spaces, and its properties have been carefully analyzed, constituting a good theoretical contribution.
- In experiments, the authors have compared not only graph neural networks but also a important family of graph kernels and the graph edit distance, representing a valuable empirical contribution.

### Weaknesses

There are several concern regarding the presentation and quality of the paper.
- Although we can guess from Example 2.2, the procedure for converting graphs into the quadruples used in Definition 2.1 is never provided in this paper. Since this process is nontrivial and the outcomes may largely depend on the way of conversion, it should be carefully explained (maybe at the beginning of Section 2).
- Since Example 2.2 is an example of Definition 2.1, please also present the actual FNGW value based on this example. Otherwise, this example should be placed before Definition 2.1.
- How were the hyperparameters of each graph kernel and graph neural network tuned in the experiments?
- The vertex-edge histogram kernel should be used instead of the edge histogram kernel (EHK) because EHK does not utilize label information, leading to an unfair comparison with other methods that utilize vertex label information.
- I believe edge label information is ignored in the WL kernel, although it is feasible to incorporate it by extending the WL, for instance, by integrating neighboring edge labels along with vertex labels. While such a straightforward extension may not yield optimal results, it is worthwhile to compare to establish a baseline WL that incorporates both vertex and edge features.
- Please include a runtime comparison, particularly with other methods in Table 1, and compare with Graph Edit Distance (GED) in Figure 4. Although the proposal might not be the fastest, showing its runtime alongside other methods adds significant value for readers of TMLR.

Minor points:
- In "Notations", $\Sigma\_n$ should be not like $\\{A, B\\}$ but $\\{A \mid B\\}$ or $\\{A : B\\}$ to distinguish enumeration and predicates.
- In Equation (1), $\\{F, A, E\\}$ should be $(F, A, B)$ as it is not a set.

---

> ### Author Response · Authors · 2024-05-24
> **Response to Reviewer ZFM9**
>
> We would like to thank the reviewer for their comments and suggestions. We did our best to answer and revise the paper consequently.
>
> > Although we can guess from Example 2.2, the procedure for converting graphs into the quadruples used in Definition 2.1 is never provided in this paper. Since this process is nontrivial and the outcomes may largely depend on the way of conversion, it should be carefully explained (maybe at the beginning of Section 2).
>
> > Since Example 2.2 is an example of Definition 2.1, please also present the actual FNGW value based on this example. Otherwise, this example should be placed before Definition 2.1.
>
>
> **Response:** Thanks for your suggestion. At the beginning of Section 2 of the revised paper, we provide a separate definition (Def. 2.1) of the quadruple, followed directly by a concrete example (Example 2.2) where we detail one possible procedure to convert the edge-featured graphs in Figure 1 into the quadruple. Then we give the definition of FNGW distance in the discrete case, followed by the actual FNGW value based on Example 2.2 (see blue text on p.3).
>
> > How were the hyperparameters of each graph kernel and graph neural network tuned in the experiments?
>
> **Response:** We elaborated on the selection of hyperparameters for both graph kernels and GNNs in Appendix C.1. This clarification has been emphasized in Section 4.1.1. (See p.8 in blue).
>
>
>
> > The vertex-edge histogram kernel should be used instead of the edge histogram kernel (EHK) because EHK does not utilize label information, leading to an unfair comparison with other methods that utilize vertex label information.
>
> **Response:** You are correct, EHK does not utilize label information. We include the results of EHK simply to demonstrate that the edge feature itself can be beneficial for the classification task. As you suggested, we have implemented
> the Vertex-edge Histogram (VEH) kernel based on GraKeL package and included its performance in the revised version. Please refer to Table 1 for details. **We will make a pull request of this implementation to GraKeL package in the future.**
>
> > I believe edge label information is ignored in the WL kernel, although it is feasible to incorporate it by extending the WL, for instance, by integrating neighboring edge labels along with vertex labels. While such a straightforward extension may not yield optimal results, it is worthwhile to compare to establish a baseline WL that incorporates both vertex and edge features.
>
>
> **Response:** Indeed, we utilized the Weisfeiler-Lehman subtree kernel (i.e., Weisfeiler-Lehman kernel with Vertex Histogram as the base kernel), which does not consider edge label information. Please refer to Table 1 in the revised paper for additional results of WLK-VEH (Weisfeiler-Lehman kernel with Vertex-edge Histogram as the base kernel), which incorporates both vertex and edge features.
>
>
> > Please include a runtime comparison, particularly with other methods in Table 1, and compare with Graph Edit Distance (GED) in Figure 4. Although the proposal might not be the fastest, showing its runtime alongside other methods adds significant value for readers of TMLR.
>
> **Response:** We acknowledge the importance of runtime considerations for TMLR readers. In the revised version, we have included the runtime of Gram computation of the graph kernels, FGW and FNGW for the classification (please refer to Table 6). However, GED is not a baseline of FNGW in structured prediction, as we employ it as a metric to assess the quality of predicted graphs in the context of a structured prediction task, where FNGW serves as the learning loss. We also provide a runtime comparison of GED and FNGW (please refer to Table 9).
>
>
> > Minor points:
> In "Notations", $\Sigma_n$ should be not like $\{A, B\}$ but $\{A \mid B\}$ or $\{A : B\}$ to distinguish enumeration and predicates. In Equation (1), $\{F, A, E\}$ should be $(F, A, B)$ as it is not a set.
>
> **Response:** We have corrected the typos mentioned: see the blue changes in the revised version. Thank you for pointing them out.

---

### Decision · Action_Editor_1vYS · 2024-06-18

**Recommendation:** Accept as is

**Comment:**

This manuscript considers the problem of supervised learning on graph-valued data. Here a common approach is to first define a metric between graphs, and then proceed with learning procedures that can leverage this notion of distance (or similarity) between the training inputs. A recent innovation in this space is to use ideas from optimal transport, such as the Wasserstein distance and extensions thereof, to formalize and compute this notion of distance between graphs. However, existing tools are not natively equipped to handle the case where both nodes and edges in the graphs have attributes. The authors address this problem by defining a new graph metric that can handle this case and demonstrate the power of their proposed methodology in a series of well-designed empirical studies on real-world datasets.

The reviewers were universally positive in their initial assessment of this work, and this sentiment was only strengthened throughout the author-reviewer discussion period. Ultimately the reviewers agreed that the paper satisfies both TMLR acceptance criteria and were unified in their recommendation of acceptance. I agree with their assessment.

I encourage the authors to reflect on the reviewer feedback and author-reviewer discussion in preparing the camera-ready version of their manuscript.

**Audience:**

Yes, the central topic of this paper -- graph learning -- plays a central roles in many areas of modern machine learning, and there is no question that a significant number of individuals in the TMLR audience would be interested in the findings of this paper.

**Claims And Evidence:**

Yes, all reviewers agree that the claims in this manuscript are supported by accurate, convincing, and clear evidence. The reviewers in particular praised the problem setting and motivation thereof, the design of the proposed methodology, and the design and findings of the empirical study.